# Impaired migration and lung invasion of human melanoma by a novel small molecule targeting the transmembrane domain of death receptor p75$^{NTR}$

Vanessa Lopes-Rodrigues [1,12], Samuel A Nyantakyi [2], Xueqing Lun[3], Xueyan Han [4,5], Jianbo Zhang[6], Ajeena Ramanujan [1,13], Shuhailah Salim [1], Michael Saleeb[2,14], Liane Babes[6], Angela Z Chou[7], Lingyu Du [7], Siyi Dong[7,8], James J Chou [7], Donna L Senger [3,6,9] & Carlos F Ibáñez [1,2,5,10,11 ✉]

## Abstract

Receptor transmembrane domains (TMDs) are crucially involved in relaying ligand information from extracellular to intracellular spaces and represent attractive targets for small molecule manipulation of receptor function. Screening a library of over 8000 drug-like compounds with an assay based on the TMD of death receptor p75$^{NTR}$, we identified a novel small molecule capable of inhibiting p75$^{NTR}$-mediated migration of human melanoma cells. Employing medicinal chemistry, a more potent derivative termed Np75-4A22 was identified that blocked nerve growth factor (NGF)-mediated melanoma invasion at sub-micromolar concentrations. The specific interaction of Np75-4A22 with the p75$^{NTR}$ TMD was confirmed by 2D NMR. Mechanistically, Np75-4A22 was found to antagonize NGF-mediated recruitment of the actin-bundling protein fascin to p75$^{NTR}$, fascin association with the actin cytoskeleton and filopodia formation. Importantly, preclinical assessment of Np75-4A22 showed high oral bioavailability, low toxicity, and significant inhibition of melanoma lung invasion in mice. These results support further development of this approach as an alternative or complementary strategy for melanoma cancer patients that do not respond to conventional chemotherapy or immune checkpoint inhibitors.

**Keywords** AraTM; Chemical Biology; Cytoskeleton; Lung Tumors; Medicinal Chemistry
**Subject Categories** Cancer; Pharmacology & Drug Discovery; Skin

## Introduction

Melanoma is the most serious form of skin cancer, and its worldwide incidence has been rising rapidly (Ferlay et al, 2019). In 2020, there were an estimated 325,000 new cases of melanoma worldwide, accounting for 1.7% of all global cancer diagnoses and 57,000 deaths (Saginala et al, 2021). This high mortality rate has been associated with the highly invasive and metastatic nature of melanoma (Saginala et al, 2021). In patients with metastatic melanoma carrying mutated forms of BRAF, which comprise about 40–50% of all melanoma patients (Davies 2002), tumor regression and prolonged overall survival can be achieved with vemurafenib and dabrafenib, two selective BRAF inhibitors (Chapman et al, 2011). However, resistance and disease progression has been observed in virtually every patient treated with these inhibitors (Chapman et al, 2011). Although strategies that target melanoma with mutant KIT have been effective, activating mutations in KIT are only present in a small percentage of melanoma patients (Pham et al, 2016). Thus, over 50% of melanoma lack a clear target for therapeutic intervention, and in those that do, acquired resistance is an ensuing problem. Currently, immune checkpoint inhibitors (ICI) are a primary treatment option available to patients with metastatic melanoma, and while these treatments provide long-term survival in some patients, recent clinical trials with ICIs indicate that over 50% of metastatic melanoma patients fail to respond to mono-therapy with either Nivolumab or Ipilimumab (targeting PD-1 and CTAL-4, respectively) (Rotte, 2019) and objective or complete response rates are in the order of 60% and 20%, respectively (Seidel et al, 2018; Kim et al, 2007) when used in combination. Why almost 40% of these patients do not respond to the very best combination of available immune therapy is not understood.

[1]Department of Physiology and Life Sciences Institute, National University of Singapore, Singapore 117456, Singapore. [2]Department of Neuroscience, Karolinska Institute, Stockholm 17177, Sweden. [3]Arnie Charbonneau Cancer Institute, Department of Oncology, Cumming School of Medicine, University of Calgary, Calgary T2N 1N4, Canada. [4]College of Biological Sciences, China Agricultural University, 100193 Beijing, China. [5]Chinese Institute for Brain Research, Zhongguancun Life Science Park, 102206 Beijing, China. [6]Lady Davis Institute for Medical Research, Jewish General Hospital, Montreal, QC, Canada. [7]Interdisciplinary Research Center on Biology and Chemistry, Shanghai Institute of Organic Chemistry, Chinese Academy of Sciences, 201203 Shanghai, China. [8]University of Chinese Academy of Sciences, 100049 Beijing, China. [9]Gerald Bronfman Department of Oncology, McGill University, Montreal, QC, Canada. [10]Peking University School of Life Sciences, PKU-IDG/McGovern Institute for Brain Research, Peking-Tsinghua Center for Life Sciences, 100871 Beijing, China. [11]Stellenbosch Institute for Advanced Study, Wallenberg Research Centre at Stellenbosch University, Stellenbosch 7600, South Africa. [12]Present address: University of Porto, Porto, Portugal. [13]Present address: MD Anderson Cancer Centre, Houston, TX, USA. [14]Present address: Nottingham Trent University, Nottingham, UK. ✉E-mail: carlos.ibanez@pku.edu.cn

The p75 neurotrophin receptor (p75[NTR], also known as NGFR, TNFRSF16 and CD271) is a member of the death receptor superfamily, characterized by the presence of a 6-helix globular domain in their intracellular region known as the "death domain" (Liepinsh et al, 1997; Vilar, 2017; Yuan et al, 2019). Prototypical death receptors include the Tumor Necrosis Factor Receptor 1 (TNFR1) and the Fas receptor (CD95). In addition to inducing apoptotic cell death in subpopulations of neurons, glia and cancer cells (Ibáñez and Simi, 2012; Friedman, 2010; Underwood and Coulson, 2008), p75[NTR] has been shown to regulate cell motility and migration in response to neurotrophins, such as nerve growth factor (NGF) (Yamauchi et al, 2004; Shonukan et al, 2003; Bentley and Lee, 2000) and is expressed at high levels in the neural crest and its cellular derivatives, including melanocytes. p75[NTR] is present in deeply invasive melanoma lesions (Shonukan et al, 2003) and has been associated with tumor cell plasticity (Boiko et al, 2010; Civenni et al, 2011; Restivo et al, 2017), progression (Radke et al, 2017), and metastases (Redmer, 2017; Radke et al, 2017) Numerous studies have highlighted the ability of p75[NTR] to regulate the survival and migration of melanoma cells in response to neurotrophins (Herrmann et al, 1993; Walch et al, 1999; Marchetti et al, 2004; Shonukan et al, 2003; Truzzi et al, 2008; Kasemeier-Kulesa and Kulesa, 2018; Restivo et al, 2017) and have implicated p75[NTR] in resistance to targeted BRAF and MEK inhibition (Rambow et al, 2018; Lehraiki et al, 2015) and immune based therapies (Landsberg et al, 2012; Mehta et al, 2018; Boshuizen et al, 2020). Together, these studies highlight the functional role of p75[NTR] and its potential as a therapeutic target for interventions aimed at mitigating metastatic melanoma.

One mechanism by which p75[NTR] becomes activated by endogenous neurotrophin ligands involves a conformational rearrangement of pre-formed receptor dimers that is transferred from extracellular to intracellular domains through a highly conserved transmembrane Cys residue (Cys[256] in human p75[NTR]) (Vilar et al, 2009). This Cys residue forms an intramembrane disulphide bridge that stabilizes constitutive dimers at the plasma membrane (Vilar et al, 2009). Neurotrophin binding to p75[NTR] induces a conformational change measurable by fluorescence resonance energy transfer (FRET) anisotropy that results in rapid but transient separation of receptor intracellular domains in an oscillatory fashion. Replacement of the transmembrane Cys to Alanine abolishes this conformational change and blunts downstream signaling in response to neurotrophins without affecting receptor expression at the cell surface. Knock-in mice carrying this replacement show significant protection from epileptic-induced neuronal damage (Tanaka et al, 2016) as well as neuropathology and memory impairment caused by overexpression of the 5xFAD Alzheimer's disease transgene (Yi et al, 2021). The crucial role played by TMD-TMD interactions in the mechanism of p75[NTR] activation suggests that targeting the receptor TMDs may represent a novel approach to develop small molecule modulators of p75[NTR]. Interestingly, the p75[NTR] TMD is fairly unique: the highest scoring protein in the mammalian genome, olfactory receptor 52E4, has less than 30% sequence identity in this region, supporting this TMD as a suitable drug target. In a recent proof-of-principle study using a small library of 1580 molecules, our laboratory adapted a screening assay based on the AraTM transcription factor system in bacteria (Russ and Engelman, 1999; Su et al, 2003) to identify a small molecule, the chalcone flavonoid NSC49652, that interacted

directly with the p75[NTR] TMD and induced profound conformational changes and activity in the full-length receptor in mammalian cells (Goh et al, 2018). The compound triggered apoptotic cell death that was dependent on p75[NTR] and JNK activity in neurons and melanoma cells, and inhibited melanoma tumor growth (Goh et al, 2018).

In the present study, we used the AraTM assay to screen a much larger and chemically diverse library of over 8,000 compounds with drug-like properties for novel small molecules targeting the p75[NTR] TMD and capable of affecting receptor function. We report the discovery of a novel molecule that was improved by medicinal chemistry to generate a derivative with good oral bioavailability, low toxicity, and the ability to block melanoma motility and migration in vitro and tissue invasion in vivo.

# Results

## A novel pyrazine interacts with the p75[NTR] TMD and induces dynamic changes in the full length receptor in mammalian cells

The AraTM assay (Russ and Engelman, 1999; Su et al, 2003) was used to screen a library of ~8000 compounds from the Chemical Biology Consortium Sweden (CBCS) for molecules capable of interacting with the p75[NTR] TMD as previously described (Goh et al, 2018). Unlike our previous pilot study, which used a small collection (≈1500 compounds) of known drugs from the National Cancer Institute (USA), the CBCS library was curated to represent a much larger chemical diversity, and to contain novel compounds that have lead-like to drug-like properties with respect to parameters such as molecular weight, hydrogen bond accepting and donating groups, lipophilicity and polar surface area. Briefly, a fusion protein consisting of the maltose binding protein (MBP) at the N-terminus, followed by the p75[NTR] TMD and the AraC bacterial transcription factor at the C-terminus was expressed in the AS19 strain of $E.\ coli$ which lacks lipopolysaccharide (LPS) in the cell wall, thereby increasing the access of different types of drugs and macromolecules to targets in the plasma membrane and cytoplasm (Good et al, 2000). TMD-mediated dimerization of AraC leads to its activation and induction of a GFP reporter expressed from a second plasmid carrying AraC binding sites in its promoter region. Small molecules may reduce or enhance TMD-TMD interactions, leading to decreased or increased GFP readouts, respectively. Compounds that affect bacterial growth (i.e. reduce $OD_{630}$ by more than 30%) are considered toxic and removed from the screen. Compounds showing a statistically significant ($P < 0.05$) change in the normalized GFP readout greater than 50% were retained in the screen and subjected to secondary and tertiary screens for reduced toxicity (i.e. no or low effects on $OD_{630}$ reading) and increased potency (high $GFP/OD_{630}$ ratio) and specificity, assessed by absence of effects on unrelated TMDs such as that from α2β3 integrin, as performed in our previous study (Goh et al, 2018) (Fig. 1A). This led to the identification of a previously undescribed pyrazine that we termed Div17E5. The molecule consists of a pyrazin-2-yl phenolic moiety joined at position 6 by a 3,4-dihalogenated anilino substituent (Fig. 1B). In dose-response studies, Div17E5 displayed an $IC_{50}$ of 12 μM in the AraTM assay and showed good selectivity for the p75[NTR] TMD as

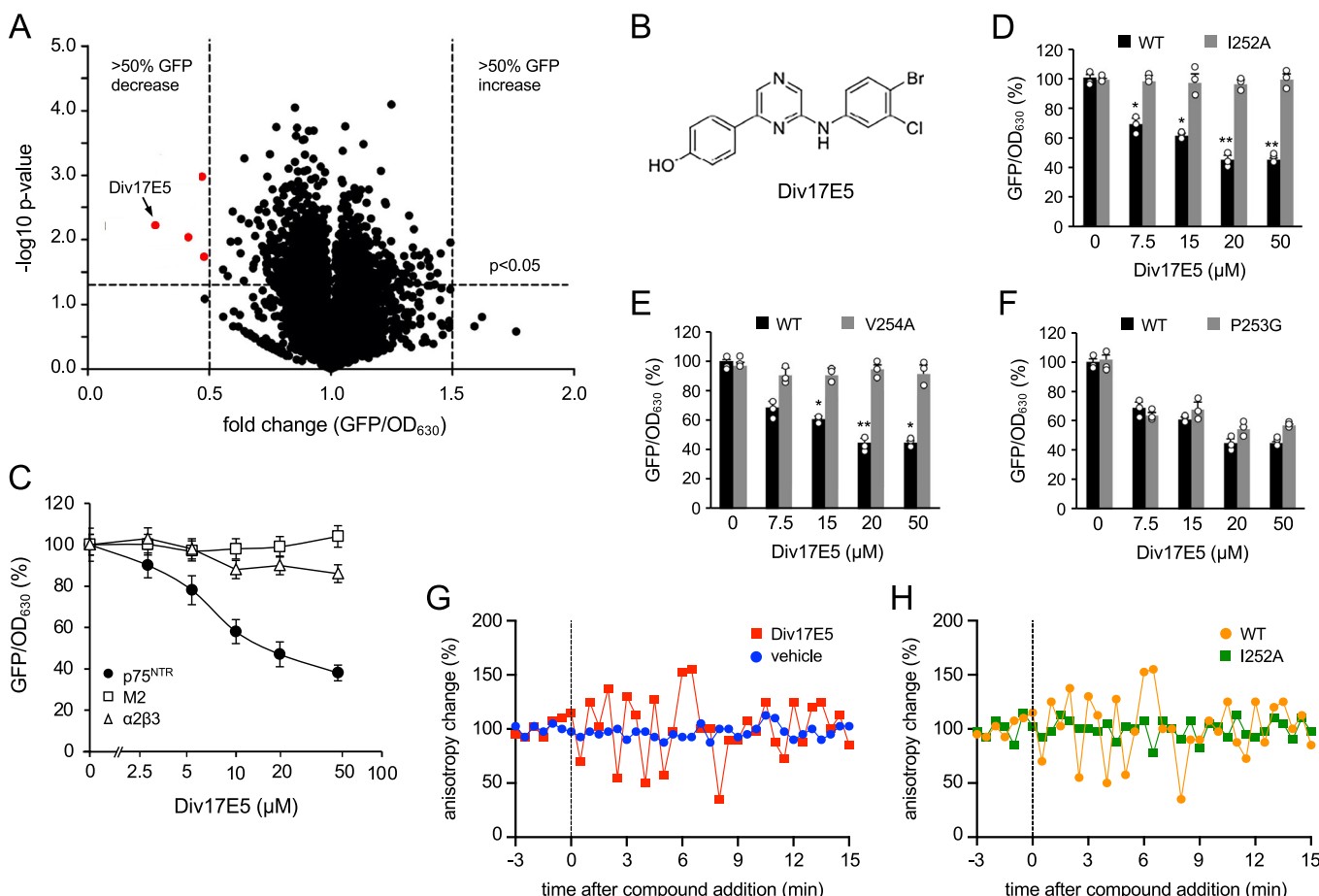

**Figure 1. Novel pyrazine Div17E5 interacts with the p75^NTR TMD and induces dynamic changes in the full length receptor in mammalian cells.**

(A) Volcano plot of AraTM screening assay of Screening Set v2010 (8482 compounds) from the Chemical Biology Consortium Sweden (www.cbcs.se). Fold change was calculated against vehicle (DMSO) and hits were defined as compounds resulting in greater than ±0.5 fold change in $GFP/OD_{630}$ signal without affecting $OD_{630}$ by greater than 0.3-fold across three independent runs (red dots). P values were calculated by one-way ANOVA ($N = 3$). Compound Div17E5 is indicated. (B) Chemical structure of Div17E5. (C) Dose response of Div17E5 in the AraTM assay of p75^NTR TMD in comparison with unrelated TMDs from α2β3 integrin and Matrix-2 protein (M2) from the viral envelope of influenza A virus. Results are plotted as mean ± SD ($N = 3$). (D–F) Comparison of wild-type (WT) p75^NTR TMD and I252A, P253G and V254A mutants in the AraTM assay in response to increasing doses of Div17E5. $GFP/OD_{630}$ signal for WT TMD without any drug was set at 100% and all other measurements are relative to that. Results are plotted as mean ± SD ($N = 3$). *$P = 0.03$; **$P = 0.006$ (one-way ANOVA followed by Tukey's multiple comparisons). (G) Live cell homo-FRET anisotropy in response to Div17E5 of full-length, wild-type human p75^NTR expressed in COS cells. Shown are representative traces of average anisotropy change after addition of Div17E5 (10 μM) or vehicle. (H) Live cell homo-FRET anisotropy in response to Div17E5 of full-length, wild-type human p75^NTR in comparison to I252A p75^NTR TMD mutant. Shown are representative traces of average anisotropy change after addition of Div17E5 (10 μM). Source data are available online for this figure.

compared to TMDs of two unrelated proteins (Fig. 1C). Mutation of TMD residues Ile^252 or Val^254 abolished the ability of Div17E5 to affect the p75^NTR TMD (Fig. 1D,E), suggesting they are key residues for drug binding. On the other hand, mutation of Pro^253 had no effect (Fig. 1F). None of the mutations diminished baseline interaction between TMDs in the absence of drug (Fig. 1D–F), indicating that Div17E5 interacted with p75^NTR TMD residues that do not normally contribute to TMD dimerization. The fact that point mutations cancelled the effects of Div17E5 in the AraTM assay suggests that the compound targeted the TMD and not the technology of the assay.

Previous real-time homo-FRET anisotropy studies by ours as well as other laboratories have shown that the intracellular death domains of p75^NTR are associated with each other (high FRET, low anisotropy state) under basal conditions (Vilar et al, 2009; Sykes

et al, 2012), but neurotrophin binding induces transient separation of p75^NTR death domains (low FRET, high anisotropy state) manifested as oscillations in real-time anisotropy measurements (Lin et al, 2015; Tanaka et al, 2016; Goh et al, 2018). COS cells expressing full length, GFP-tagged rat p75^NTR were treated with vehicle or Div17E5 and changes in anisotropy levels were recorded over time. The compound induced oscillations in real-time anisotropy (Fig. 1G), similar to those induced by endogenous p75^NTR ligands, such as NGF (Lin et al, 2015; Tanaka et al, 2016; Goh et al, 2018) In agreement with the AraTM data, mutation of transmembrane residue Ile^253 (equivalent to human Ile^252) to Ala blunted homo-FRET changes induced by Div17E5 (Fig. 1H). Together, these findings suggest that Div17E5 binding to the TMD of p75^NTR induces conformational changes in the full-length receptor that can be propagated to intracellular death domains.

## Div17E5 induces apoptosis of human melanoma cells in a p75[NTR]-dependent manner

One of the main biological readouts of p75[NTR] activity in response to neurotrophins is the induction of caspase-mediated apoptotic cell death (Underwood and Coulson, 2008; Friedman, 2010; Ibáñez and Simi, 2012). We investigated whether Div17E5 displayed p75[NTR]-mediated pro-apoptotic activity in the human melanoma cell line A875, which expresses high levels of this receptor (Grob et al, 1983), and a derived line in which p75[NTR] expression had been abolished by shRNA knock-down (A875-NT and A875-shp75, respectively, Fig. 2A). Div17E5 treatment induced apoptosis, as indicated by cleavage of Poly (ADP-ribose) Polymerase (PARP), in control A875-NT cells, while p75[NTR] knock-down significantly reduced the pro-apoptotic effects of the compound (Fig. 2B). Apoptosis induced by Div17E5 could be blocked by the pan-caspase inhibitor Q-VD-pOH (Fig. 2C). In line with these findings, Div17E5 decreased viability of A875 melanoma cells in a dose-dependent manner, while A875 cells lacking p75[NTR] were largely refractory to the compound at doses up to 10 µM (Fig. 2D). Above this concentration, both cell lines were sensitive to the compound, indicating in vitro off-target effects at high concentrations, which is not unusual for target-specific drugs.

## Div17E5 impairs cell motility and chemotaxis of human A875 melanoma cells through targeting p75[NTR] but independent of its pro-apoptotic effects

As p75[NTR] is known to regulate motility and migration in melanoma cells, we tested whether Div17E5 affected these activities. Using the wound-healing assay in cell monolayers, we found that Div17E5 significantly impaired motility of A875-NT cells (Fig. 3A,B). Importantly, the effect of Div17E5 on cell motility was not due to apoptosis, as it could not be rescued by treatment with the caspase inhibitor Q-VD-pOH (Fig. 3A,B). Div17E5 did not significantly affect wound closure in A875-shp75 cells (Fig. 3C), suggesting that its effects on cell motility are mediated through p75[NTR].

These results led us to hypothesize that Div17E5 may antagonize NGF-induced chemotaxis of A875 melanoma cells, which has previously been shown to be mediated by p75[NTR] (Iwamoto et al, 1996; Walch et al, 1999). We tested this in trans-well Boyden chambers with NGF in the lower compartment by assessing the extent of cell migration across the membrane in the presence and absence of DivE175 and caspase inhibitor. A875-NT cells responded robustly to NGF, while the knock-down cells did not, although they still responded to serum (Fig. 3D,E). Div17E5 did not significantly affect background cell migration in the absence of NGF but strongly inhibited NGF-induced chemotaxis of A875-NT cells (Fig. 3D). This effect could not be rescued by the caspase inhibitor, indicating that it was not due to apoptosis (Fig. 3D). Together these results demonstrate that Div17E5 can interfere with cell motility and chemotaxis of human A875 melanoma cells through p75[NTR].

## Medicinal chemistry defines structural requirements of Div17E5 activity and identifies a more potent analog

We applied medicinal chemistry to investigate the molecular determinants underlying Div17E5 activity and to identify more potent and better tolerated compounds. We synthesized 31 chemically diverse analogs (Series 1) carrying different groups linked to either of the two benzene rings (Fig. 4A). R1 and R2 groups ranged from electron-donating groups, electron-withdrawing groups, and bulky hydrophobic groups in different combinations, all the while maintaining the -NH-linker at position 6 (Series 1, compounds MS_11 to MS_71, MS_100 and MS_114 in Fig. EV1). To understand the essence of the amino (-NH-) linker (Fig. 4B), 6 additional analogs lacking it were also synthesized (Series 2A, MS_72 to MS_78 in Fig. EV1). Finally, a carbon linker was introduced at position 2 of the pyrazine in MS_63 (Fig. 4C) to generate an additional 4 compounds (Series 2B, MS_109 to MS_112 in Fig. EV1). All 6 compounds from Series 2A were inactive (Fig. EV2), stressing the importance of the amino group for the activity of Div17E5. Neither did the 4 compounds in Series 2B result in improved activity (Fig. EV3). From these studies, only two analogs from Series 1 were retained, namely MS_18 and MS_63, which displayed similar IC$_{50}$s as Div17E5 in the AraTM assay but with somewhat lower bacterial toxicity, as assessed by OD$_{630}$ measurements (Fig. EV4). MS_18 and MS_63 featured electron-donating 4-methoxy and 3-dimethylamino, respectively, as the R1 group, and electron-withdrawing 4-nitro and 4-bromo-2,3-dichloro motifs, respectively, as the R2 group. However, when tested in the wound healing assay in A875-NT cells, Div17E5 was still the better of the three (Fig. 4D).

MS_18 and MS_63 were then further modified to yield Series 3A/B and 4A/B, respectively (Figs. EV5, 6, 7 and 8). The design strategy for analogs in Series 3A was to maintain the 4-nitro (4-NO$_2$) substituent on ring C found in MS_18, while modifying substitutions on ring A (Fig. 4E). Conversely, for 3B, we maintained the 4-methoxy (4-OCH$_3$) substituent on ring A found in MS_18, while modifying substitutions on ring C. The same design principles were employed for Series 4A and 4B on MS_63 (Fig. 4F). In total, 144 new compounds were synthesized and tested in the AraTM assay. A detailed analysis of structure–activity relationships for compounds in Series 3A/B and 4 A/B is presented separately (see "Methods"). From these efforts, Np75-4A22 was retained (Fig. 5A), as it showed better performance at low micromolar concentrations and improved IC$_{50}$ compared to Div17E5 (Fig. 5B). The activity of this compound in the AraTM assay was further characterized using mutant p75[NTR] TMDs. Similar to Div17E5, Np75-4A22 was affected by replacement of Ile$^{252}$ and unaffected by mutation of Pro$^{253}$ (Fig. 5C,D). However, unlike Div17E5, Np75-4A22 was unaffected by mutation of Val$^{254}$ (Fig. 5E), indicating that the two compounds interact with a partially overlapping but distinct set of residues in the p75[NTR] TMD.

Chemical shift perturbations derived from 2D nuclear magnetic resonance (NMR) experiments confirmed the specific interaction of Np75-4A22 with the p75[NTR] TMD (Fig. 5F–H). Significant chemical shifts were detected in a number of TMD residues upon addition of the compound. Interestingly, assignment of the cross-peaks validated Ile$^{252}$ as one of the key residues undergoing conformational change upon Np75-4A22 binding (Fig. 5F), in agreement with the mutagenesis data (Fig. 1D). Importantly, no or minimal chemical shift perturbations were observed upon addition of the compound to unrelated TMDs, such as those of the TrkB receptor tyrosine kinase (Fig. 5G) or tumor necrosis factor receptor 2 (TNFR2, Fig. 5H). Together, these results demonstrate the specific interaction of the compound with the p75[NTR] TMD and identify several TMD residues that are either directly involved in drug binding or significantly affected by it.

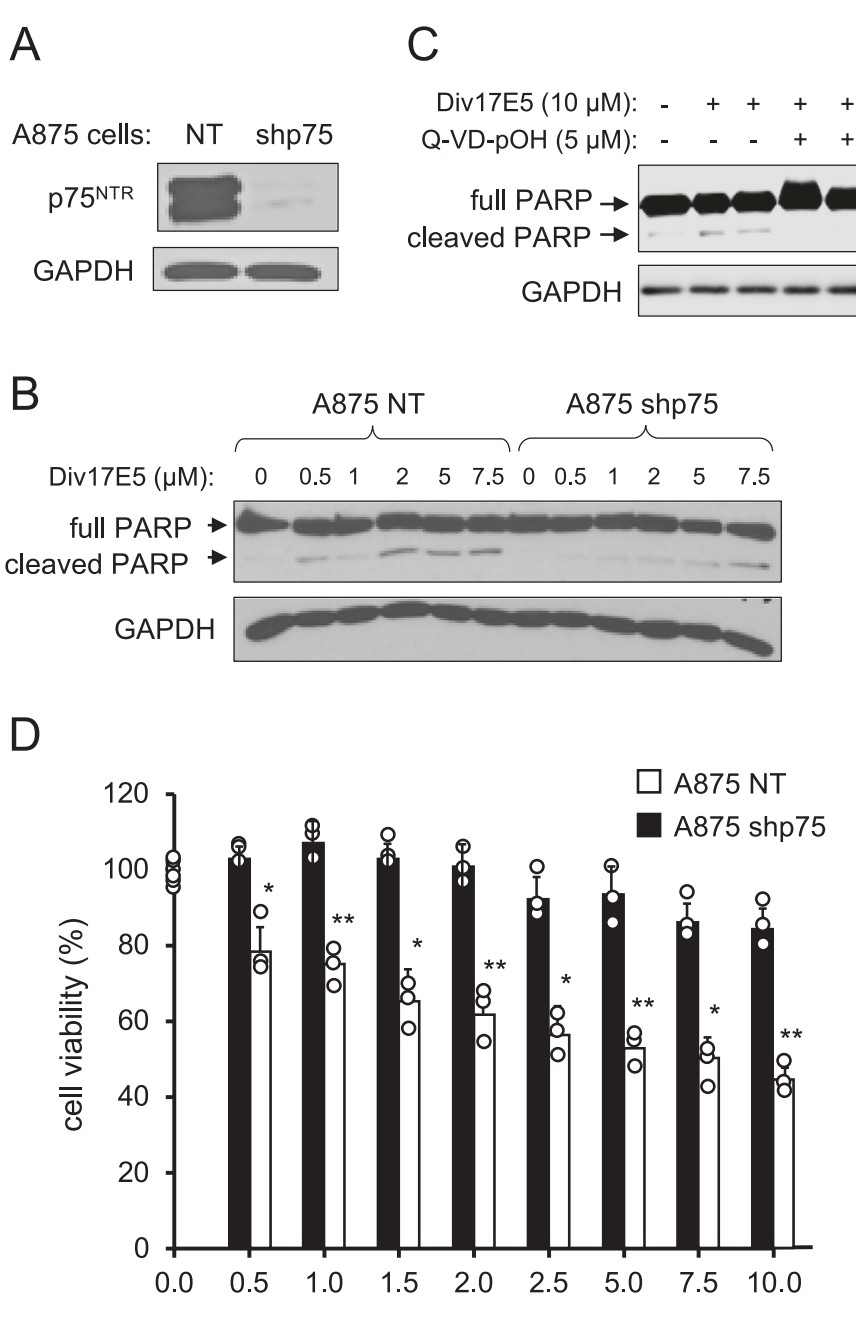

**Figure 2. Div17E5 induces apoptosis of human melanoma cells in a p75^NTR-dependent manner.**

(A) Expression of p75^NTR in A875 control (NT, non-targeting) and knock-down (shp75) cells. (B) Dose response analysis of cleaved PARP induction by Div17E5 in A875 control (NT) and knock-down (shp75) cells. (C) Western blot analysis of cleaved PARP in A875 melanoma cells in response to Div17E5 (10 μM) in the presence or absence of pan-caspase inhibitor Q-VD-pOH (5 μM). Reprobing for GAPDH was used as loading control. The experiment was repeated three times with comparable results. (D) Dose-dependent cell viability of A875 control (NT) and knock-down (shp75) cells in response to Div17E5. Results are plotted as mean ± SD ($N = 3$). *$P = 0.04$; **$P = 0.006$ (one-way ANOVA followed by Tukey's multiple comparisons). Source data are available online for this figure.

## Np75-4A22 inhibits melanoma cell invasion at submicromolar concentrations without inducing cell death

In contrast to Div17E5, Np75-4A22 did not induce apoptosis of A875 melanoma cells in the submicromolar and low micromolar range, as assessed by PARP cleavage (Fig. 6A). However, Np75-4A22 was significantly more potent than Div17E5 in the wound healing assay, displaying considerable inhibitory activity at low micromolar and submicromolar concentrations, at which Div17E5 was totally inactive (Fig. 6B). In the Boyden chamber assay of cell chemotaxis, 1 μM Np75-4A22 blocked all NGF activity, while

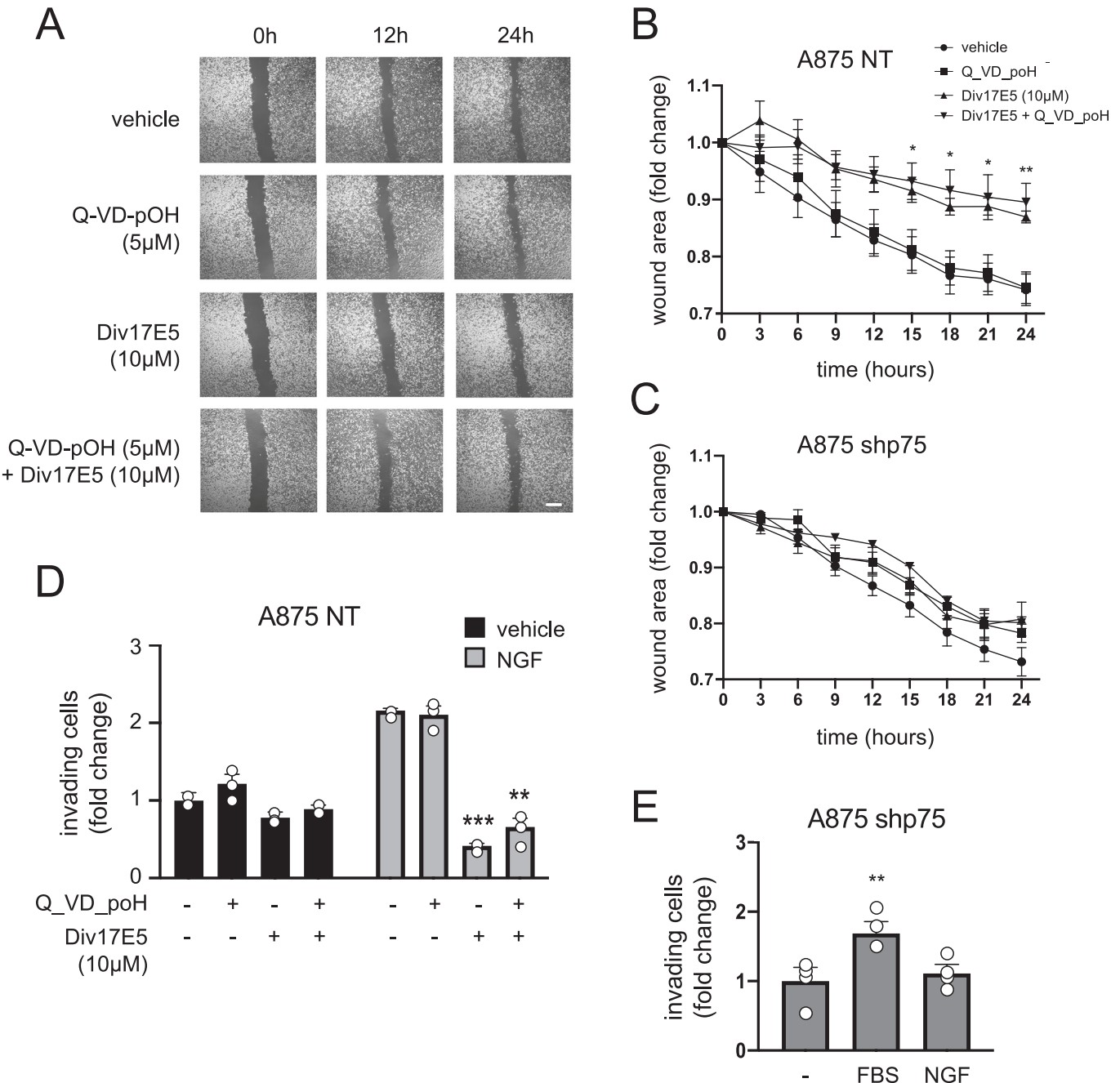

**Figure 3. Div17E5 impairs cell motility and chemotaxis of A875 melanoma cells through p75^NTR.**

(A) Phase contrast photomicrographs of wound healing cell motility assay in A875 cells treated with vehicle or Div17E5 in the presence or absence of pan-caspase inhibitor Q-VD-pOH. Size bar (bottom right), 2 mm. (B, C) Time course of wound healing in A875-NT control (B) and knock-down shp75 (C) cells in response to Div17E5 in the presence or absence of Q-VD-pOH. Results are plotted as mean ± SD ($N = 3$). *$P = 0.044$; **$P = 0.008$ (one-way ANOVA followed by Tukey's multiple comparisons). (D) Boyden chamber analysis of A875 cell chemotaxis in response to NGF in the presence or absence or Div17E5 or pan-caspase inhibitor Q-VD-pOH. Results are plotted as mean ± SD ($N = 3$). **$P = 0.004$; ***$P = 0.00076$ (one-way ANOVA followed by Tukey's multiple comparisons). (E) Boyden chamber analysis of cell chemotaxis of knock-down (shp75) A875 cells in response to NGF or serum (FBS). Results are plotted as mean ± SD ($N = 3$). **$P = 0.008$ (one-way ANOVA followed by Tukey's multiple comparisons). Source data are available online for this figure.

Div17E5 was inactive at that concentration (Fig. 6C). Interestingly, Np75-4A22 was also capable of significantly reducing NGF-induced A875 chemotaxis at submicromolar concentrations (Fig. 6D). WM115 is a tumorigenic primary melanoma human cell line derived from a primary epitheloid tumor showing

moderate levels of p75^NTR expression. NGF induced the migration of these cells to the lower side of Boyden chambers. Importantly, 1 μM Np75-4A22 blocked this activity without affecting the spontaneous migration of the cells (Fig. 6E). Thus, the Np75-4A22 analog was less toxic than Div17E5 but considerably more

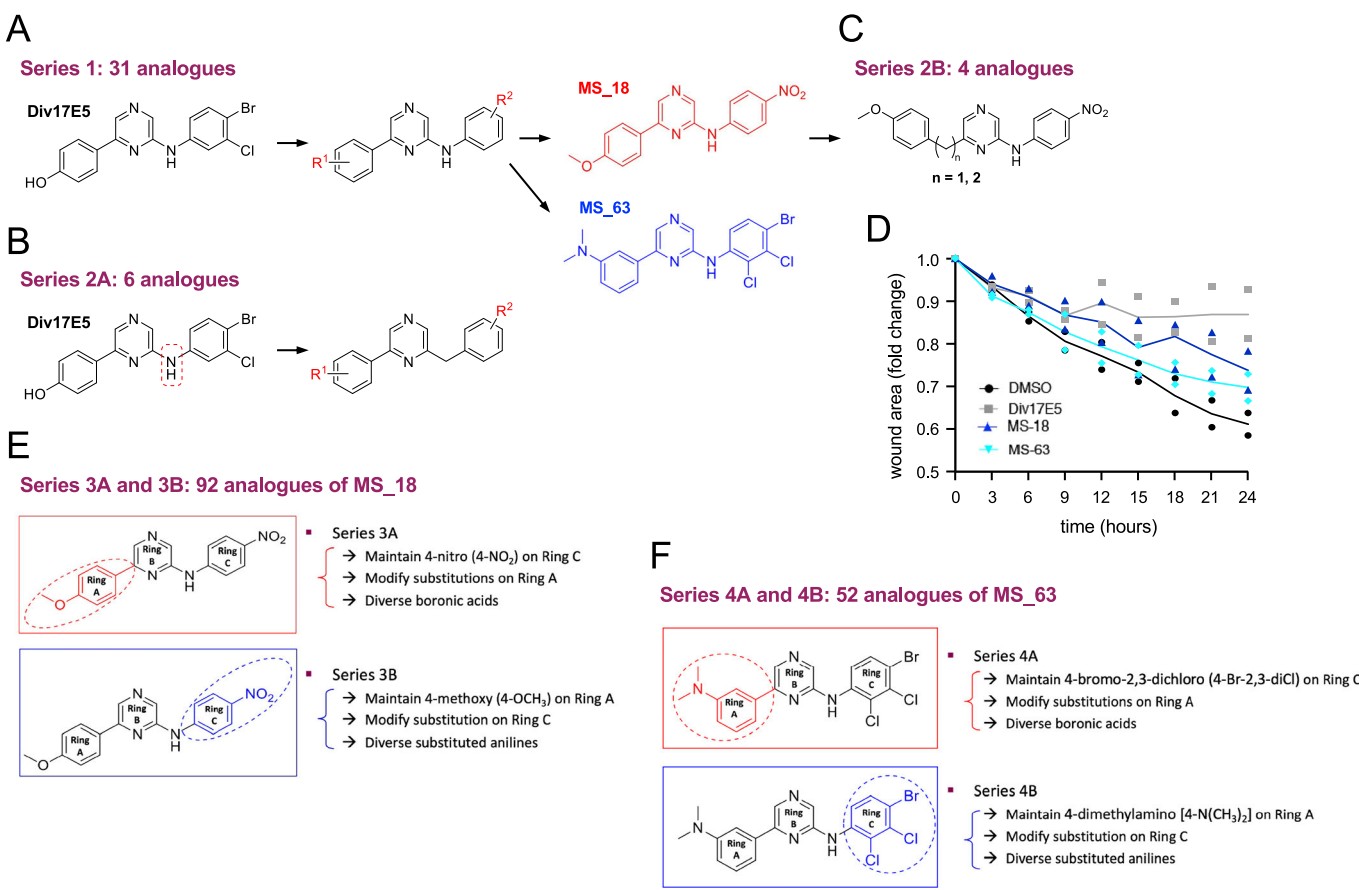

**Figure 4. Medicinal chemistry strategy for derivation of Div17E5 analogs.**

(A) Series 1 included 31 chemically diverse analogs carrying different groups linked to either of the two benzene rings (R1 and R2). Chemical structures of MS_18 and MS_63, the two compounds retained from this series, are shown. (B) Series 2B included 6 analogs lacking the amino (–NH–) linker with. R1 and R2 groups. (C) Series 2C included 4 analogs carrying an extra carbon linker at position 2 of the pyrazine in MS_63. (D) Time course of wound healing in A875 NT cells in response to 10 μM Div17E5, MS_18 or MS_63. Results are plotted as mean ± SD (N = 2). (E) Series 3A and 3B included 92 analogs of MS_18 that maintained the 4-nitro (4-NO2) substituent on ring C while modifying substitutions on ring A (3A), or the 4-methoxy (4-OCH3) substituent on ring A while modifying substitutions on ring C (3B). (F) Series 4A and 4B included 52 analogs of MS_63 using the same principles as Series 3A and 3B, respectively. Source data are available online for this figure.

potent at blocking A875 melanoma cell invasion. It was also able to inhibit NGF-induced cell migration in WM115 cells, suggesting a broad ability to prevent human melanoma invasion.

## Np75-4A22 blocks NGF-mediated recruitment of fascin to p75^NTR, fascin association with the cytoskeleton and filopodia formation in A875 melanoma cells

It has been shown that the actin-bundling protein fascin interacts with p75[NTR] in an NGF-dependent manner (Shonukan et al, 2003). NGF also enhanced the association of fascin with the actin cytoskeleton in melanoma cells, as shown by increased fascin levels in detergent-insoluble fractions upon NGF treatment (Shonukan et al, 2003). Importantly, the same study also showed that NGF-induced migration of A875 melanoma cells was inhibited by a mutant of fascin that precludes its binding to actin. Together, these findings indicated that one mechanism by which NGF enhances migration of melanoma cells is through fascin recruitment to

p75[NTR] as well as increased fascin association with the actin cytoskeleton. We investigated whether Np75-4A22 has any effects on the ability of NGF to regulate fascin in A875 melanoma cells. Fascin co-immunoprecipitated with p75[NTR] in control NT cells but not in shp75 knockdown cells (Fig. 7A). In agreement with the previous study, NGF treatment strongly stimulated the association of fascin with p75[NTR] (Fig. 7A). Interestingly, treatment with Np75-4A22 at 1 μM completely suppressed the effect of NGF reducing fascin/p75[NTR] interaction to baseline levels (Fig. 7A). Moreover, while NGF increased the levels of fascin in the detergent insoluble fraction of A875-NT cells, a proxy for fascin association with the actin cytoskeleton, 1 μM Np75-4A22 also blocked this effect (Fig. 7B). In order to assess direct changes in the actin cytoskeleton, we stained cultures of A875 cells with fluophor-conjugated phalloidin to label actin filaments (Fig. 7C). A significant increase in the number of filopodia per cell was observed upon NGF treatment of A875 NT cells (Fig. 7D) which was not detected in A875 shp75 cells (Fig. 7E), indicating NGF-

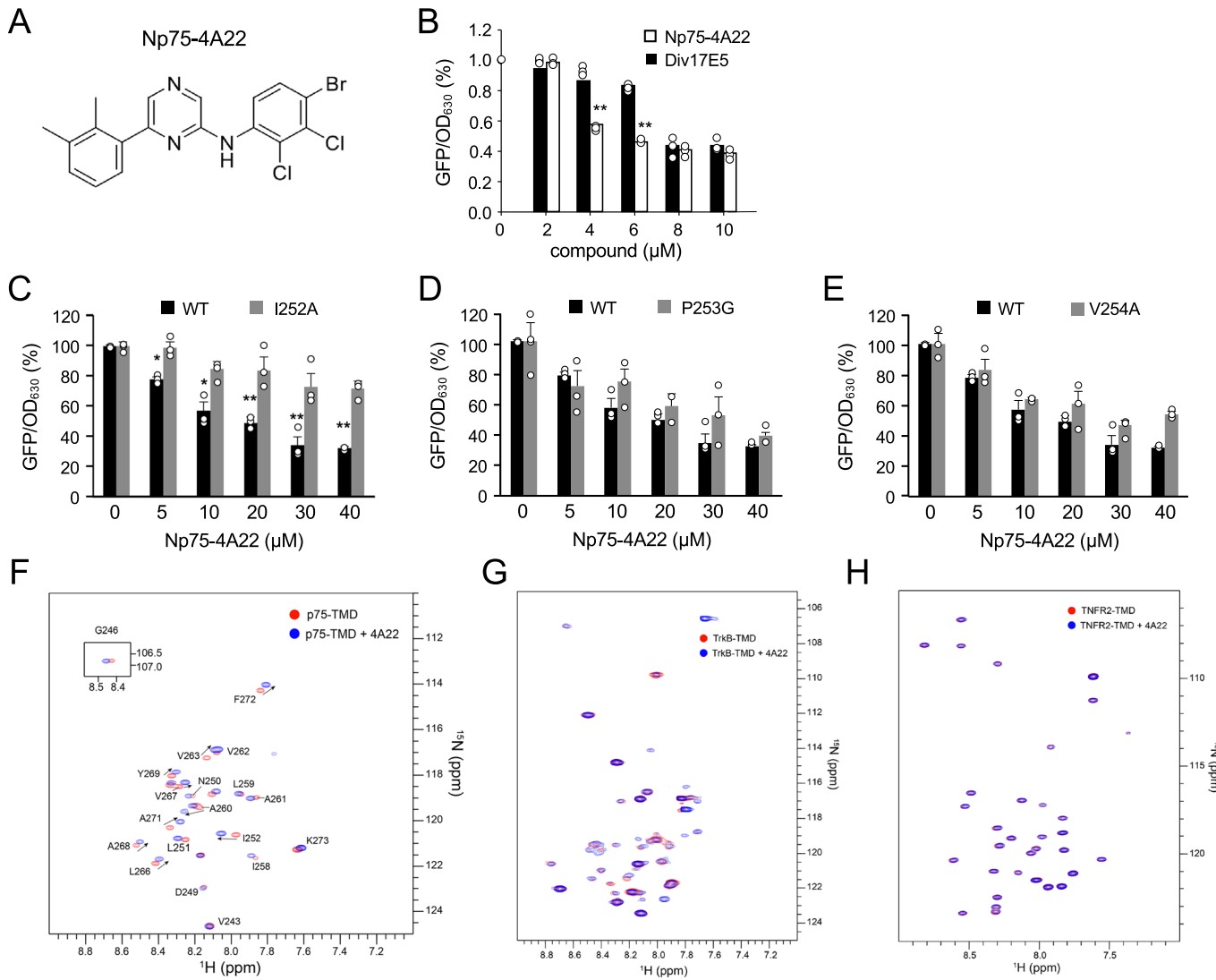

**Figure 5. Np75-4A22, a more potent analog of Div17E5 identified by medicinal chemistry.**

(A) Chemical structure of Np75-4A22. (B) Dose response of Div17E5 and Np75-4A22 in the AraTM assay of p75$^{NTR}$ TMD. Results are plotted as mean ± SD ($N = 3$). **$P = 0.007$ (one-way ANOVA followed by Tukey's multiple comparisons). (C–E) Comparison of wild-type p75$^{NTR}$ TMD and I252A, P253G and V254A mutants in the AraTM assay in response to increasing doses of Np75-4A22. GFP/OD$_{630}$ signal without any drug added was set at 100%. Results are plotted as mean ± SD ($N = 3$). *$P = 0.035$; **$P = 0.008$ (one-way ANOVA followed by Tukey's multiple comparisons). (F–H) Specific interaction between Np75-4A22 (4 mM) and p75$^{NTR}$ TMD (1 mM) in bicelles that mimic a lipid bilayer. (F) $^1$H-$^{15}$N TROSY-HSQC spectra of p75$^{NTR}$ TMD in DMPC-DH6PC bicelles ($q = 0.5$) in the absence (red) and presence (blue) of 4 molar excess Np75-4A22. Cross-peaks undergoing significant chemical shift are indicated with arrows and the corresponding residues labeled. (G) The same as (F) for the TMD of TrkB receptor tyrosine kinase, showing mostly non-specific interaction. (H) Same as (F) for the TMD of TNF receptor 2 (TNFR2), showing essentially no interaction. Source data are available online for this figure.

induced effects on the actin cytoskeleton that were mediated by p75$^{NTR}$. Np75-4A22, at either 1 or 5 μM, had no effects per se in either cell line but, importantly, completely abolished the effects of NGF on filopodia of A875 NT cells (Fig. 7D). These results indicated that, mechanistically, Np75-4A22 may inhibit melanoma cell migration and invasion by interfering with NGF-mediated recruitment of fascin to the receptor, fascin association with the actin cytoskeleton and filopodia formation. Together, our in vitro studies prompted us to investigate whether Np75-4A22 may be able to impair melanoma migration and tissue invasion in vivo.

## Oral administration of Np75-4A22 reduces the spread and seeding of melanoma cells in the lungs of a mouse xenograft model

To determine optimal dosing of Np75-4A22, a pharmacokinetic study was performed following single intravenous (3 mg/kg) or oral gavage (10 mg/kg) administration in female and male C57Bl/6 mice. Plasma concentration-time profiles and main pharmacokinetic parameters are summarized in Fig. EV9A,B, respectively. Briefly, Np75-4A22 exhibited low clearance and long half-life following intravenous (i.v.) administration, as well as rapid

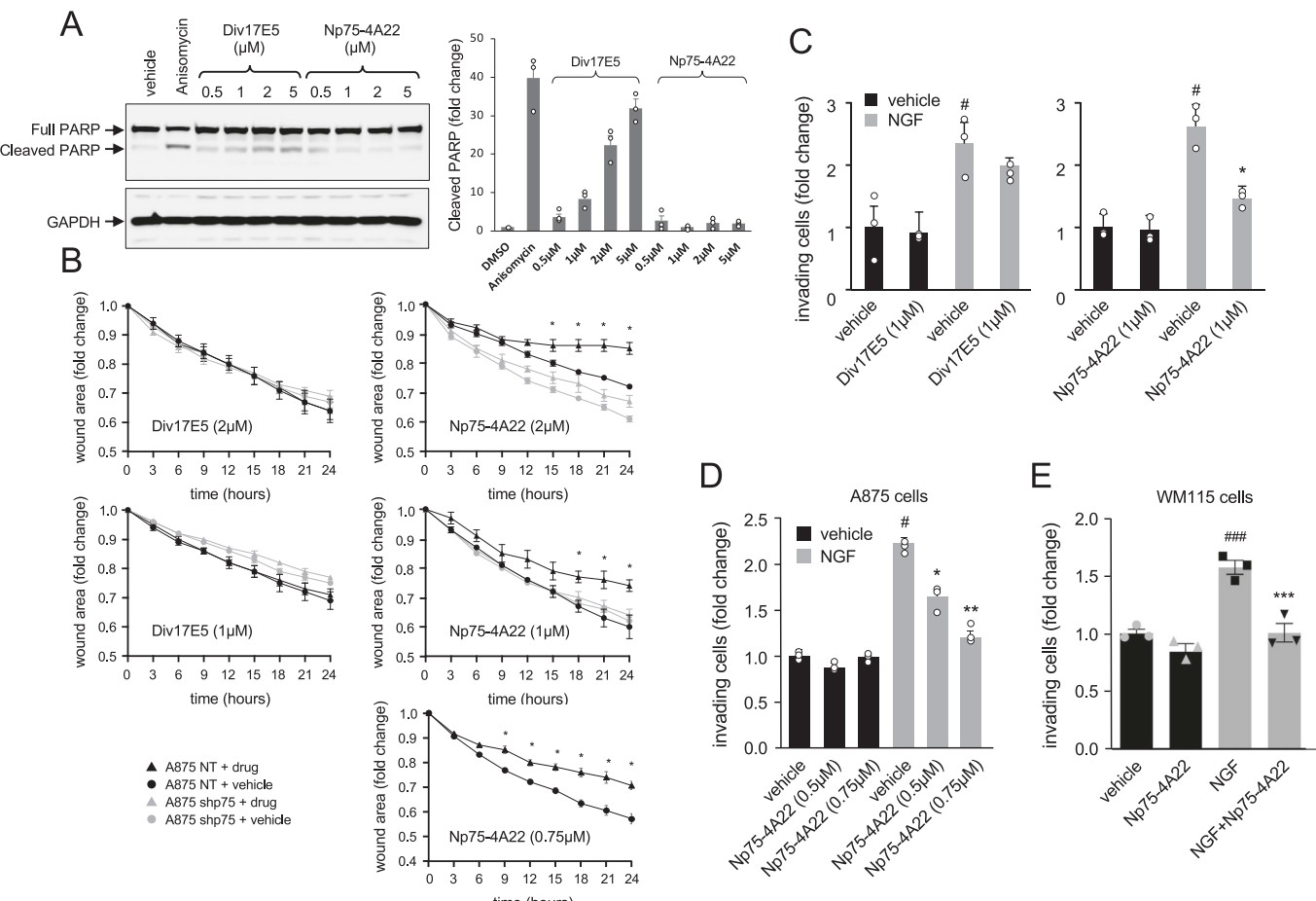

**Figure 6. Np75-4A22 inhibits melanoma cell invasion at submicromolar concentrations without inducing cell death.**

(A) Western blot analysis of cleaved PARP in A875 melanoma cells in response to increasing concentrations (0.5 to 5 μM) of Div17E5 and Np75-4A22. A potent inducer of apoptosis, anisomycin (1 μM), was used as positive control. Reprobing for GAPDH was used as loading control. The histogram on the right shows quantification expressed as average of three independent experiments, each performed in triplicate ± SEM. (B) Time course of wound healing in A875 control (NT) and knock-down (shp75) cells in response to low micromolar and submicromolar concentrations of Div17E5 (left side) or Np75-4A22 (right side). Results are plotted as mean ± SEM ($N = 3$). *$P = 0.035$ (one-way ANOVA followed by Tukey's multiple comparisons). (C) Boyden chamber analysis of A875 cell chemotaxis in response to NGF (50 ng/ml) in the presence or absence or Div17E5 (left side graph) or Np75-4A22 (right side graph). Results are plotted as mean ± SEM ($N = 3$ independent experiments each performed in triplicate). #$P = 0.0465$ vs. no NGF; *$P = 0.0435$ vs. no drug (one-way ANOVA followed by Tukey's multiple comparisons). (D) Boyden chamber analysis of A875 cell chemotaxis in response to NGF (50 ng/ml) in the presence of submicromolar concentrations of Np75-4A22. Results are plotted as mean ± SEM ($N = 2$ independent experiments each performed in triplicate). #$P = 0.0477$ vs. no NGF; *$P = 0.0462$ and **$P = 0.0078$ vs. NGF without drug (one-way ANOVA followed by Tukey's multiple comparisons). (E) Boyden chamber analysis of WM115 cell chemotaxis in response to NGF (100 ng/ml) in the presence of Np75-4A22 (1 μM). Results are plotted as mean ± SEM ($N = 3$ independent experiments each performed in triplicate). ###$P = 0.0004$ vs. vehicle; ***$P = 0.0004$ vs. NGF without drug (one-way ANOVA followed by Tukey's multiple comparisons). Source data are available online for this figure.

absorption and good oral bioavailability following oral administration in both male and female mice. To establish maximal dosing for in vivo studies, toxicology analyses were performed on female and male C57Bl/6 mice 7 days after a single dose oral administration of Np75-4A22 at 50, 100 or 200 mg/kg. No mortality or abnormal clinical signs were observed or treatment-related effects on body weight, food consumption or gross pathology in either male or female C57BL/6 mice treated with up to 200 mg/kg Np75-4A22. Subsequently, a repeated dose tolerability study was performed to determine toxicity of Np75-4A22 after daily oral administration at 200 mg/kg/day to female and male C57Bl/6 mice for a period of 21 consecutive days. No mortality was observed in either females or males, and no treatment-related effects were

observed in males, while a mild decrease in lymphocyte cellularity and increase granulopoiesis were observed in females. In addition, and prior to in vivo preclinical melanoma studies, Np75-4A22 was administered by oral gavage at 200 mg/kg daily (5 days on, 2 days off) for 3 weeks in immunodeficient female SCID mice. No mortality or adverse clinical signs, including body weight, food consumption or effects on gross histological pathology assessed on formalin-fixed paraffin-embedded lung, liver, kidney or brain were observed in any of the Np75-4A22 treated mice.

Next, to assess the potential inhibitory effects of Np75-4A22 on melanoma cell migration and tissue invasion in vivo, we utilized the highly metastatic human A875 xenograft model. As our data indicated that Np75-A422 was not killing the tumor cells but

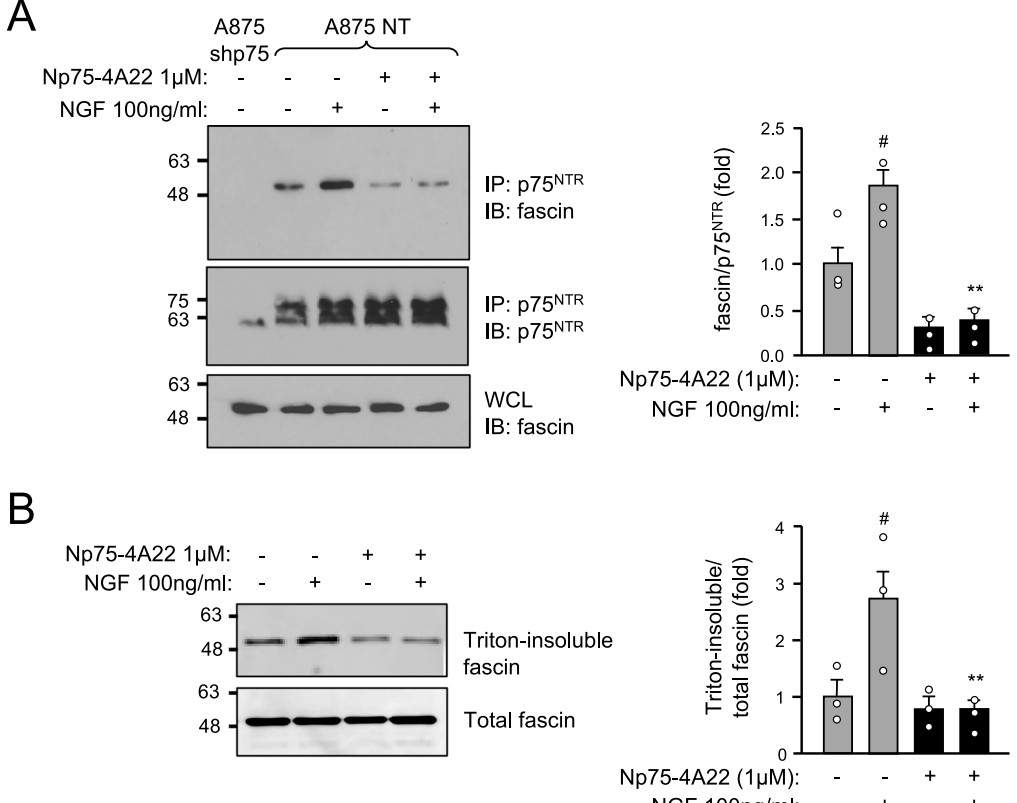

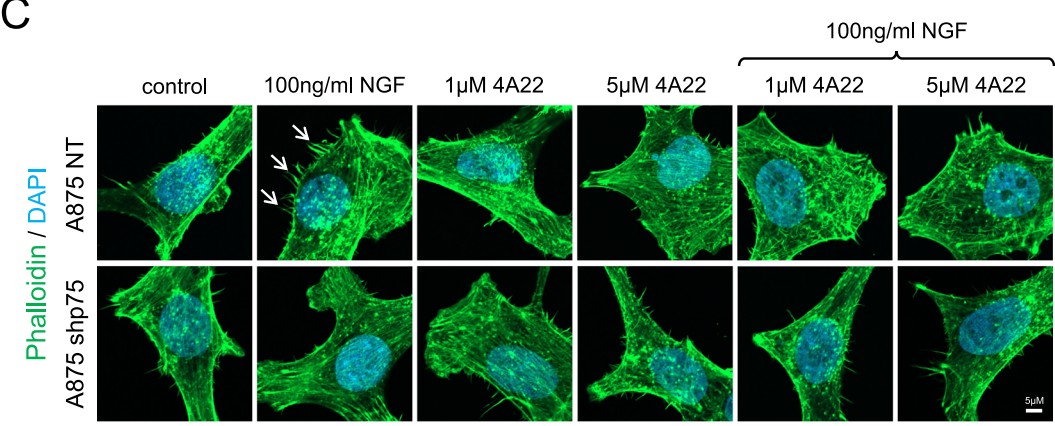

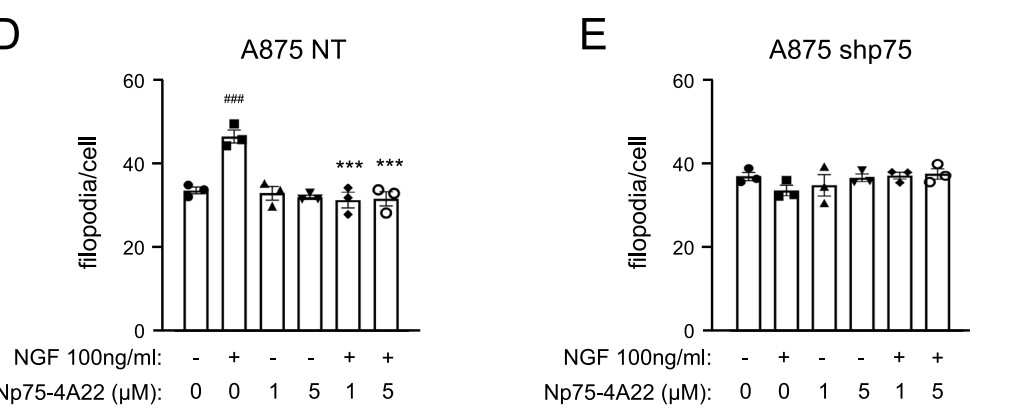

**Figure 7. Np75-4A22 blocks NGF-mediated recruitment of fascin to p75[NTR], NGF-induced fascin association with the cytoskeleton and NGF-stimulated filopodia formation.**

(A) Representative Western blot analysis (IB) of fascin in p75[NTR] immunoprecipitates (IP) and whole cell lysates (WCL) of A875 melanoma cells after overnight treatment with NGF or Np75-4A22 as indicated. The histogram on the right shows quantification expressed as average of three independent experiments, each performed in triplicate ± SD. #$P = 0.0091$ vs. control (first bar); **$P = 0.0088$ vs. NGF alone (second bar) (one-way ANOVA followed by Tukey's multiple comparisons). (B) Representative Western blot analysis of fascin in Triton-insoluble extracts of A875 melanoma cells after overnight treatment with NGF or Np75-4A22 as indicated. The histogram on the right shows quantification expressed as average of three independent experiments, each performed in triplicate ± SD. #$P = 0.0412$ vs. control (first bar); **$P = 0.0082$ vs. NGF alone (second bar) (one-way ANOVA followed by Tukey's multiple comparisons). (C) Photomicrographs of cultured A875 control (NT) and knock-down (shp75) cells treated with NGF or Np75-4A22 as indicated and stained with fluorophor-conjugated phalloidin (green), to reveal actin filaments, and DAPI (blue). White arrows denote some of the filopodia in NGF-treated cells. Scale bar, 5 μM. (D, E) Quantification of filopodia per cell in A875 NT (D) and A875 shp75 (E) cells treated with 100 ng/ml NGF or 1 or 5 μM Np75-4A22 as indicated. Average of three independent experiments, each performed in triplicate, are shown ± SEM. ###$P = 0.0004$ vs. vehicle (first bar); ***$P = 0.0001$ vs. NGF alone (second bar). Source data are available online for this figure.

inhibiting invasion and migration, we first assessed the effect of Np75-A422 on seeding and metastatic spread of tumor cells to the lung. Initially, to mimic metastatic dissemination from the primary tumor site, A875-NT or A875-shp75 cells expressing GFP-Luc (for live-monitoring of tumor growth) were injected intravenously in female SCID mice 24 h following a single oral administration of Np75-A422 (200 mg/kg) or drug vehicle alone. Mice received 3 cycles of Np75-A422 (200 mg/kg/day) following a 5 day on and 2 days off drug schedule for a total of 3 weeks (Fig. 8A). Metastatic spread to the lungs was monitored weekly by luciferase activity using the Xenogen IVIS-200 Optical in vivo luciferase imaging system and mice were sacrificed after the third week. Treatment with Np75-4A22 resulted in a dramatic reduction of lung tumors in the mice that received A875 melanoma cells expressing p75[NTR] (Fig. 8B). In addition, assessment of the metastatic spread was also quantified by histological assessment of non-consecutive lung sections stained for human nucleolin to detect the tumor cells. As shown in Fig. 8C,D, a marked reduction of p75[NTR]-positive lung tumors (both cancer cells as well as cancer lesions) was observed in lungs of mice treated with Np75-4A22 compared to vehicle alone. In addition, and as expected, high expression of p75[NTR] was detected in lung tumors established by A875-NT cells but was negligible in tumors derived from A875-shp75 cells (Fig. EV10). Consistent with the requirement of p75[NTR] for melanoma migration and invasion, we found that the loss of p75[NTR] expression in A875 melanoma cells by shRNA (A875-shp75) results in a significant decrease in lung melanoma cells when compared to p75[NTR]-positive A875 (A875-NT) (Fig. 8C,D), a finding consistent with previous studies on the effects of p75[NTR] in melanoma cell dispersion (Iwamoto et al, 1996). Together, these results show that treatment in vivo with Np75-4A22 can substantially decrease the spread and tissue invasion of melanoma cells in mice. Further, we found that p75[NTR]-negative tumors (A875-shp75), although fewer and smaller than those generated by A875-NT cells, were not significantly affected by Np75-4A22 (Fig. 8C,D) thus supporting the target specificity of Np75-4A22 for p75[NTR].

Finally, we investigated whether delayed administration of Np75-4A22 was able to affect engraftment and growth of metastatic tumors in the lung. To this end, drug treatment (orally at 200 mg/kg) was initiated 3 weeks after intravenous injection of A875-NT or A875-shp75 cells in female SCID mice and continued for additionally 4 weekly cycles (5 days on/2 days off) as above (Fig. 8E). Metastatic spread to the lungs was monitored weekly by bioluminescent imaging of luciferase-expressing melanoma cells. Treatment with Np75-4A22 had a very robust effect, reducing

A875-NT lung tumors to very low levels, comparable to those attained by A875-shp75 cells lacking p75[NTR], essentially cancelling the effects of the receptor in A875-NT cells (Fig. 8F). We conclude that Np75-4A22 can be efficacious at reducing the spread and seeding of melanoma cells in the lung when administered either before or after cell implantation.

## Discussion

When detected early, melanoma is highly treatable, however if left undetected it can spread and metastasize to other areas in the body with organotrophic selectivity to the lungs and brain (Nguyen et al, 2009). Herein we describe the identification of a novel small molecule targeting the TMD of p75[NTR], a receptor implicated in the development of invasive (Shonukan et al, 2003) and metastatic (Radke et al, 2017; Redmer, 2017) melanoma. This compound was found to inhibit melanoma cell migration and chemotaxis, and the generation of a more potent derivative by medicinal chemistry significantly reduced melanoma invasion to the lungs. Importantly, Np75-4A22 was able to inhibit the cell migration in two different human melanoma cell lines. Div17E5 and Np75-4A22 are likely to directly interact with the p75[NTR] TMD as (i) their effects in the AraTM assay were abolished by point mutations in specific TMD residues and (ii) Np75-4A22 induced chemical shift perturbations in the TMD of p75[NTR], but not in that of other transmembrane receptors, as assessed by 2D NMR. Interestingly, while mutation of either Ile[252] and Val[254] affected Div17E5 interaction with the p75[NTR] TMD, only mutation in the former interfered with Np75-4A22 activity, indicating that the two compounds interact with partially overlapping but distinct TMD residues. Also notable is the observation that neither compound was affected by mutation of Pro[253], a substitution we previously showed to interfere with the activity of NSC49652, a compound identified in a previous study (Goh et al, 2018), again suggesting distinct modes of action of compounds targeting the p75[NTR] TMD.

To understand structure-function relationships of the drug/TMD interaction, and to improve the activity and safety profile of Div17E5, a series of derivatives (185 in total) were synthesized and studied. Series 1 and 2 analogs were initially made to understand the preferred pharmacophore for optimal binding. Modifications ranging from electron-donating groups, electron-withdrawing groups, and bulky hydrophobic and steric groups in different combinations, all the while maintaining the amino (-NH-) linker at position 6, resulted in 41 chemically diverse analogs with varying

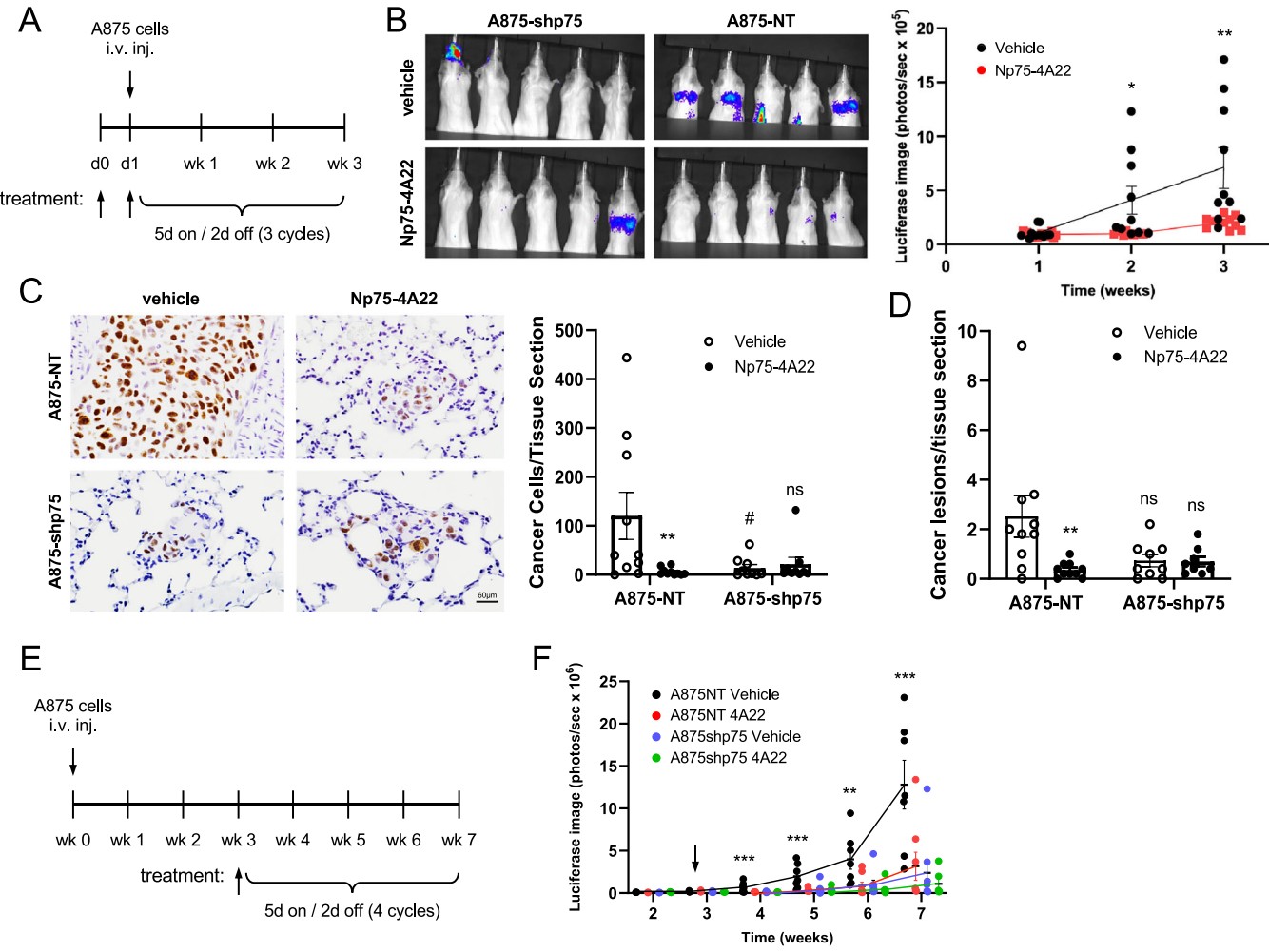

**Figure 8. Oral administration of Np5-4A22 reduces the spread and seeding of melanoma lung tumors in a mouse xenograft model.**

(A) Schematic of experimental design for the assessment of human melanoma xenograft lung invasion following treatment with Np75-4A22. (B) Live-monitoring of lung tumor growth of A875-shp75 and A875-NT melanoma cells using the Xenogen IVIS-200 luciferase imaging system. Bioluminescent imaging of luciferase-expressing melanoma cells in the lungs of SCID mice was measured 1, 2 and 3-weeks following treatment with Np75-4A22 (200 mg/kg/day) or vehicle alone. Representative images taken 3 weeks following implantation are shown (left panel). Graph shows the average quantification of luciferase signal ($N = 8$–10 mice/group; right panel). Data are expressed as average ± SEM. *$P = 0.0238$: **$P = 0.0107$; Student $t$ test. (C) Histological analysis of lung tumor burden in SCID mice bearing A875-NT or A875-shp75 tumors 3 weeks following of treatment with Np75-4A22 or vehicle alone. Left panel shows representative images of formalin-fixed paraffin-embedded lung sections stained by immunohistochemistry with a human-specific nucleolin antibody (brown) to detect the presence of tumor cells. Sections were counterstained with hematoxylin (blue). Scale bar, 60 μm. Right panel shows quantification of the number of cancer cells per lung tissue section. Five non-sequential lung sections, 100 μm apart, were counted and averaged for each mouse ($N = 8$–10 mice per group). Data is expressed as the mean ± SEM. Means that are significantly different are indicated. A875-NT vehicle vs Np75-4A22 **$P = 0.0179$; A875-NT vs A875-shp75 #$P = 0.0367$; A857-shp75 vehicle vs Np75-4A22 not significant (n.s.), one-way ANOVA with Tukey's multiple comparisons. (D) Quantitation of Human Nucleolin positive lesions (metastatic nodules). For each individual mouse, the number of lesions from 5 non-consecutive tissue sections were quantified. Data are expressed as average ± SEM. **$P = 0.0117$; ns, not statistically significant; $N = 9$ or 10 mice/group; one-way ANOVA with Tukey's multiple comparisons. (E) Schematic of experimental design for the assessment of human melanoma xenograft lung invasion model after delayed treatment with Np75-4A22 commencing 3 weeks after cell implantation. (F) Quantification of bioluminescent imaging of luciferase-expressing A875-NT and A875-shp75 melanoma cells in the lungs of SCID mice measured weekly after cell implantation as indicated. Treatment with Np75-4A22 (200 mg/kg/day) or vehicle was initiated 3 weeks after cell implantation (arrow). Data are expressed as the mean ± SEM. Means that are significantly different between treatments as analyzed by ANOVA are indicated. **$P = 0.0111$ (6 weeks); ***$P = 0.0034$ (4 weeks), 0.0012 (5 weeks), 0.0056 (7 weeks) (A875NT vehicle vs. A875NT 4A22). $N = 8$–10 mice/group, one-way ANOVA followed by Tukey's multiple comparisons. Source data are available online for this figure.

activity and toxicity profiles. This led to identification of MS_18 and MS_63 with similar IC50s as Div17E5 but lower potency in the wound healing assay. Consistent with Div17E5, electron-donating aromatic groups were preferred at position 2 of the pyrazine, with electron-withdrawing anilino motif favorable at position 6. Further, to understand the essence of the amino linker, 6 analogs, in which this was removed, were synthesized. All 6 compounds showed no activity, confirming the importance of the amino group for TMD binding. The NH-group, which can act both as a H-bond donor and acceptor, may participate in hydrogen bonding with key amino acids (e.g., Ile[252]) in the TMD binding pocket thereby contributing essential binding activity. MS_18 and MS_63 were further modified

to yield 144 additional compounds in Series 3A/B and 4A/B. The best compounds identified from this collection were Np75-3A21 and Np75-4A22. Both compounds carry electron-donating groups attached to the phenyl ring at position 2 of the pyrazine core (ring A), as well as electron-withdrawing motifs were attached to the anilino substituent at position 6 (ring C). The fact that these hits were representatives of Series 3A and 4A, while none of the modifications attempted in Series 3B and 4B yielded a better alternative, indicates that electron-donating groups on ring A are crucial when coupled with electron-withdrawing motifs on ring C.

Div17E5 acts as a p75$^{NTR}$ agonist with respect to apoptosis induction, but as an antagonist of NGF and p75$^{NTR}$-mediated cell motility and chemotaxis. This behavior can be rationalized by the way in which p75$^{NTR}$ engages with different downstream interactors and pathways. Recent research has revealed that downstream p75$^{NTR}$ effectors compete for binding to the receptor intracellular domain (Lin et al, 2015; Kisiswa et al, 2018). The terms of such competitions are likely to be dictated by the relative abundance of downstream effectors in different cell types, the presence or absence of p75$^{NTR}$ co-receptors, the nature of the ligand engaging the receptor (e.g mature neurotrophins, pro-neurotrophins, myelin-derived ligands, etc.), as well as their relative concentrations, binding affinities and relative timing of interaction. Thus, a ligand that enhances p75$^{NTR}$ coupling to the NF-kB pathway, such as the mature form of NGF, can also reduce receptor binding to RhoGDI and coupling to the RhoA pathway (Yamashita and Tohyama, 2003). Interestingly, myelin associated glycoprotein (MAG) can tilt that balance in the opposite direction in the presence of the p75$^{NTR}$ co-receptor NgR (Wang et al, 2002; Wong et al, 2002). In cerebellar neurons, the mature and unprocessed (pro) forms of NGF antagonize each other in the induction of p75$^{NTR}$-mediated cell survival vs. cell death pathways, respectively (Kisiswa et al, 2018). Against this background, it is reasonable to propose that Div17E5 binding to the p75$^{NTR}$ TMD likely induces a particular arrangement of TMDs and overall conformation dynamics that favors one downstream pathway (i.e. apoptosis) at the expense of another (i.e. cell migration). Interestingly, Np75-4A22, an analog of Div17E5, was significantly more potent than Div17E5 at blocking melanoma cell invasion, without inducing cell death, at low micromolar and submicromolar concentrations. The two compounds are likely to interact differently with the p75$^{NTR}$ TMD, as Np75-4A22, unlike Div17E5, was not affected by mutation of Val$^{254}$, suggesting that the p75$^{NTR}$ TMD is exquisitely sensitive to small structural alterations that can bias different downstream pathways in one or another direction. Together, these results indicate that it is possible to identify small molecules targeting the p75$^{NTR}$ TMD which can elicit distinct and selective functional downstream outcomes.

Mechanistically, we found that Np75-4A22 inhibited NGF-mediated recruitment of fascin to p75$^{NTR}$, association of fascin with the cytoskeleton and filopodia formation. Fascin, an actin-bundling protein, has been shown to play a role in cell motility through its localization in filopodia and lamellipodia forming at the leading edge of motile cells in response to adhesion to different types of extracellular matrix components (Adams, 1997). Moreover, fascin overexpression in kidney epithelial cells has been shown to promote their migratory activity in Boyden chambers (Yamashiro et al, 1998). Importantly, fascin levels have been found to be upregulated in human breast cancer cell lines (Grothey et al, 2000a), and tumor progression in ovarian cancer, as well as invasion in breast cancer

were found to correlate with fascin expression levels (Grothey et al, 2000b). Our results indicate that one of the mechanisms by which Np75-4A22 is able to inhibit melanoma cell invasion involves interference with NGF-mediated recruitment of fascin to p75$^{NTR}$, fascin partitioning in the detergent insoluble fraction of melanoma cells, a proxy of its association with actin fibers, and filopodia formation. They further suggest that fascin may in part exert its role in cancer cell motility and invasion by linking p75$^{NTR}$ to the actin cytoskeleton.

Pharmacokinetic and toxicology studies indicated that Np75-4A22 displays rapid absorption, high oral bioavailability and low or no toxicity. We can not rule out at this point that the treatment with Np75-4A22 affected brain functions which standard histological pathology cannot assess, such as effects on learning or memory. Preclinical studies investigating the therapeutic efficacy of Np75-4A22 on dissemination of melanoma cells from circulation into the lung, a well-known primary metastatic site for these cancer cells (Nguyen et al, 2009), were then performed. Using a highly melanoma xenograft model, daily administration of Np75-4A22 was shown to significantly inhibit the establishment of lung tumors. Importantly, robust inhibitory effects were observed regardless of whether the drug was applied before or after implantation of melanoma cells in the animals. In addition, and consistent with this observation, knockdown of p75$^{NTR}$ in human melanoma cells resulted in a similar decrease in lung tumors with no further effect observed in the presence of Np75-4A22, supporting the specificity of the compound for p75$^{NTR}$. These results are consistent with the in vitro studies showing that Np75-4A22 effectively inhibited NGF-mediated cell motility and chemotaxis of melanoma cells in a p75$^{NTR}$-dependent manner and at submicromolar concentrations. Together these data demonstrate a role for p75$^{NTR}$ in the development of melanoma-derived lung tumors and establishes Np75-4A22, as a potential therapeutic inhibitor in the systemic dissemination of melanoma.

In summary, this study provides new evidence supporting the feasibility of targeting the TMDs of single-pass transmembrane receptors for the identification of small molecules capable of modifying receptor activity in useful ways. Np75-4A22 represents a promising lead in the development of therapies against melanoma metastasis which could prove particularly beneficial for patients showing poor responsiveness to conventional chemotherapy or immune checkpoint inhibitors.

## Methods

**Reagents and tools table**

| Reagent/resource | Reference or source | Identifier or catalog number |
|---|---|---|
| **Experimental models** | | |
| AS19 bacteria cells (*E. coli*) | Good et al, 2000 | n/a |
| COS-7 cells (green monkey) | ATCC | CRL-1651 |
| A875 sh-p75$^{NTR}$ cells (*H. sapiens*) | Goh et al, 2018 | n/a |
| A875-NT cells (*H. sapiens*) | Goh et al, 2018 | n/a |
| WM115 cells (*H. sapiens*) | ATCC | CRL-1675 |

| Reagent/resource | Reference or source | Identifier or catalog number |
|---|---|---|
| SCID mice (*M. musculus*) | Charles River | Strain Code 394 |
| **Recombinant DNA** | | |
| p75[NTR] TMD | Goh et al, 2018 | n/a |
| Ara-GFP | Goh et al, 2018 | n/a |
| p75[NTR]-EGFP* | Goh et al, 2018 | n/a |
| pMM-LR6 | Fu et al, 2019 | n/a |
| **Antibodies** | | |
| P75ntr ECD | Alomone Labs | ANT-007 |
| P75ntr ECD | Promega | G323A |
| H nucleolin | Abcam | ab136649 |
| GAPDH | Cell Signalling Technologies | 2118 |
| Actin | Cell Signalling Technologies | 4967 |
| PARP | Cell Signalling Technologies | 9542 |
| Fascin | Invitrogen | MA5-11483 |
| Alexa Fluor™ 488 Goat anti-Mouse IgG | ThermoFisher | A-11029 |
| Alexa Fluor™ 568 Goat anti-Rabbit | ThermoFisher | A-11011 |
| **Chemicals, enzymes and other reagents** | | |
| Chemical library | Chemical Biology Consortium Sweden | www.cbcs.se |
| Div17E5 | Glixx | Custom synthesis |
| Np75-4A22 | SAI Life India | Custom synthesis |
| Q-vd-Oph | Sigma-Aldrich | 1135695-98-5 |
| **Software** | | |
| nmrPipe | Delaglio et al, 1995 | n/a |
| Ccpnmr | Vranken, 2005 | n/a |
| Statistical Analysis Software | SAS Institute | n/a |
| GraphPad Prism | GraphPad Software Inc | n/a |

## Chemicals and antibodies

The library of 8442 chemically diverse compounds was obtained from the Chemical Biology Consortium Sweden (CBCS, www.cbcs.se). Bulk amounts of Div17E5 used in downstream experiments were custom synthesised by Glixx Laboratories at a purity greater than 98%. Bulk amounts of Np75-4A22 were custom synthesised by SaiLife India at a purity greater than 98%. NGF was purchased from R&D Systems. Q-vd-Oph was purchased from Sigma-Aldrich. Antibody against human p75[NTR] extracellular domain used for immunoprecipitation was obtained from Alomone Labs (#ANT-007); the one used for immunofluorescence was from Promega (#G323A). Human-specific mouse polyclonal nucleolin antibody was from Abcam (#ab136649). GAPDH, Actin and PARP antibodies were purchased from Cell Signalling Technologies. Fascin antibody (#MA5-11483) was purchased from Invitrogen.

Secondary antibodies used for immunofluorescence were Alexa Fluor™ 488 Goat anti-Mouse IgG (H + L) Highly Cross-Adsorbed Secondary Antibody (ThermoFisher # A-11029) and Alexa Fluor™ 568 Goat anti-Rabbit IgG (H + L) Cross-Adsorbed Secondary Antibody (ThermoFisher # A-11011).

## AraTM screening assay

The AraTM assay was used to assess conformational changes and binding strength in a pair of interacting TMDs (Su and Berger, 2012, 2013). The assay was performed as previously described (Goh et al, 2018). Briefly, p75[NTR] TMD cDNA (encoding NLIPVYCSI-LAAVVVGLVAYIAFKRW) was subcloned into AraTM chimera plasmid and transformed in AS19 LPS-negative *E. coli* (which allows the uptake of a variety of molecules, peptides and nuclei acids) (Good et al, 2000) along with the Ara-GFP reporter plasmid. The p75[NTR] TMD sequence used is identical in all sequenced mammalian homologs of p75[NTR], including human and mouse. An overnight culture was diluted 1:100 in fresh LB medium and allowed to grow till reached between 0.2 and 0.5 optical density (OD) 630, after which IPTG was added to a final concentration of 1 mM to induce the expression of the p75[NTR] TMD–AraTM chimera. The bacterial culture was added to a black-rim clear bottom 96-well plates (Corning #3631) pre-plated with the compounds (final concentration of 20 $\mu$M). The plates were incubated at 38 °C to allow IPTG-induced expression of the TM-AraTM chimera and after 4 h were centrifuged to pellet the bacterial cells. LB media was aspirated and replaced with Phosphate Buffered Saline (PBS), and bacteria cells were resuspended by vigorous shaking for 10 min. GFP signal was measured in each well (excitation 475 nm, emission 509 nm) and bacterial density was determined by measurement of $OD_{630}$ in a microplate plate reader (BioTek).

## Homo-FRET anisotropy microscopy

Anisotropy imaging in COS-7 cells that were transfected with a rat p75[NTR]-EGFP* fusion construct was done as previously described (Goh, Lin et al, 2018). EGFP* carries a A207K mutation rendering monomeric EGFP. Changes in anisotropy were expressed as fold change at each time point (every 30 s) in comparison to the mean of 6 time points obtained prior to addition of treatment. Images were acquired using Nikon Ti-E based live cell epi-fluorescence microscope and MetaMorph software and analyzed using MatLab from Mathworks.

## Cell culture and immunoblotting

COS7, A875 and WM115 cell lines were cultured under standard conditions in DMEM supplemented with 10% fetal calf serum, 100 units/ml penicillin, 100 mg/ml streptomycin, and 2.5 mM gluta-mine. All cells were authenticated and tested for mycoplasma contamination. A875-NT (control) and A875 shp75 (knockdown) cell lines were described previously (Goh et al, 2018). As mentioned in that paper, the PiggyBac non-targeting (NT) vector contains a shRNA insert (sequence; CAACAAGATGAAGAGCACCAA, provided by the manufacturer System Bioscience) that does not target any known genes from any species. Cell lysis and whole cell protein extraction were carried out in Lysis Buffer [50 mM Tris/HCl pH

7.5, 1 mM EDTA, 270 mM Sucrose, 1% (v/v) Triton X-100, 1 mM benzamidine, 1 mM PMSF, 0.1% (v/v) 2-mercaptoethanol, and in the presence of phosSTOP (Roche) phosphatase inhibitor cocktail mix as per manufacturer instructions]. Protein concentration was determined by Bradford Assay. Addition of Laemmli sample buffer and analysis by SDS–PAGE and Western Blot. Immunoblots were developed using the ECL Western Blotting Kit (Thermo Scientific) and exposed to Kodak X-Omat AR films.

For assessment of actin filaments, A875 cells were plated on glass coverslips in 24-well plate at a confluency of $3 \times 10^4$ cells/well. One day after plating, the cells were treated with 1 µM or 5 µM Np75-4A22 and NGF (100 ng/ml) or PBS for 24 h. Cells were fixed in 4% paraformaldehyde, permeabilized in 0.1% Triton-X-100, stained with Phalloidin-iFluor 488 (Abcam) at 1:2000 dilution and DAPI for 1 h. Coverslips were mounted and visualized in a laser confocal microscope.

For co-immunoprecipitation analysis of p75$^{NTR}$/fascin interactions, A875 sh-p75$^{NTR}$ and A875-NT cells were plated in a 10 cm tissue culture dish at a confluency of $3 \times 10^6$ cells/dish in normal growth media. Twenty-four hours after plating, the cells were treated with 1 µM 4A22 and NGF (100 ng/ml) for 12 h prior to harvest and lysis in lysis buffer (50 mM Tris/HCl pH 7.5, 1 mM EDTA, 270 mM Sucrose, 1% (v/v) Triton X-100, 0.1% (v/v) 2-mercaptoethanol and 60 mM n-Octyl-β-D glucopyranoside) containing protease inhibitor cocktail. The cellular extracts were then centrifuged at 4 °C top speed on a benchtop centrifuge for 1 min and the supernatant was collected. For immunoprecipitation, A875 cell extracts were incubated for 16 h at 4 °C on a rotating wheel with 0.8 µg of anti-p75$^{NTR}$ antibody (ANT-007, Alomone) and then incubated with Sepharose protein-G beads (GE Healthcare). The beads were collected by brief centrifugation (2 min, 2500 rpm, 4 °C), washed three times with 1 ml of Wash Buffer (with composition same as the lysis buffer). After the last wash, pelleted beads were aspirated off the wash buffer followed by addition of Laemmli sample buffer and analysis by SDS–PAGE followed by Western Blot using anti-fascin antibody (MA5-11483, Invitrogen).

For detergent-insolubility assay of fascin, A875 NT cells were treated with or without 100 ng/ml NGF and 1 µM of Np75-4A22 for 24 h. A single cell suspension was obtained by trypsinization followed by centrifugation and detergent extraction with lysis buffer (with 0.5% Triton X-100). Then, immediately centrifuged at 8700 g for 3 min. The resulting pellets (containing the Triton X-100-resistant proteins) were suspended in lysis buffer without Triton X-100. In parallel, cells were centrifuged and then directly suspended in lysis buffer. The Triton X-100- resistant proteins and total cell lysates were analyzed by SDS–PAGE and Western blotting with anti-fascin antibody.

## Cell viability, motility and chemotaxis analyses

To measure the cytotoxic effect of Div17E5, A875 NT or shp75 cells were seeded at 2000 cells per well (in 100 µL DMEM) in a 96-well plate and incubated for 12 h. Next, Div17E5 at various concentrations was added and incubated at 37 °C in 5% $CO_2$ for 2 days. Following treatment, 10 µL Presto Blue® (resazurin, ThermoFIsher) was added to each well of the 96-well plate and incubated for 2 h at 37 °C in 5% $CO_2$. Cellular fluorescence in each well was measured at 535 nm excitation and 615 nm emission using a

microtiter plate reader. Data was converted into the relative cell viability (%) from the fluorescence of cells in each treatment relative to that of the DMSO control group (set as 100%).

For wound healing assay, cells were grown until confluency and subjected to a wound across the cell mono-layer. Cells were then incubated with serum-free media either with DMSO or a small molecule (with or without 5 µM Q_VD_poH) and imaged at specific time-points. Cell migration is reflected as fold change in wound area over time.

For Boyden chamber assay, Corning® BioCoat™ Matrigel® Invasion Chambers 8.0 µm PET Membrane were used (cat. no. 354480). A875 NT or WM115 cells were serum-starved for 16 h and then seeded at 60,000 cells per well in serum-free media under different conditions, including DMSO (vehicle) or small molecules Div17E5 or Np75-4A22 at different concentrations to the top chamber of the Matrigel-coated trans-well chamber. The pan-caspase inhibitor Q_VD_poH (at 5 µM) was also introduced in the top chamber in some experiments. Two different chemoattractants, NGF (at 50 ng/ml or 100 ng/ml as indicated) or 10% FBS were added to the bottom chamber. Assays were carried out for 8 h after which filters were fixed in 4% PFA and stained with DAPI. Cells that had remained on the top of the filter were gently scrapped away with a cotton tip. The filter lower side was imaged on an inverted fluorescence microscope and counted using ImageJ software (NIH).

## Synthetic methods

### *Procedure A (Amination 1)*

2,6-Dichloropyrazine (1–2 eq) and the substituted aniline (1–2 eq) were dissolved in anhydrous N,N-dimethylformamide (DMF) and cooled to −78 °C. Sodium tert-butoxide (NaO-tBu) 2.5 M in tetrahydrofuran (THF) or sodium tert-pentoxide (NaO-t-pent) 40% in THF (3.5 eq) was added dropwise and the reaction mixture stirred and allowed to warm to room temperature. Stirring was continued for between 3 and 24 h, by monitoring the reaction via TLC and/or LCMS. On completion, the reaction mixture was dissolved in dichloromethane (DCM), washed twice with water and once with saturated brine solution. The organic portion was dried with magnesium sulfate ($MgSO_4$), filtered and concentrated, before being purified by column chromatography on a SNAP (Biotage) automated purification system using isohexane: ethyl acetate (0–100%).

### *Procedure B (Amination 2)*

This method was used for some compounds for which Procedure A failed to yield the product of interest. 2,6-dichloropyrazine (1–2 eq) and the substituted secondary amine (1–2 eq) were dissolved in anhydrous acetonitrile at room temperature. N, N-diisopropylethylamine (DIPEA, 2.5 eq) was added to the reaction mixture and stirred at 80 °C for 3-4 h. The reaction mixture was cooled to room temperature, dissolved in DCM, washed twice with water and once with saturated brine solution. The organic portion was dried with $MgSO_4$, filtered and concentrated, before being purified by column chromatography on SNAP (Biotage) automated purification system using DCM (100%).

### *Procedure C (Suzuki coupling 1)*

Substituted 6-chloro-pyrazine-2-amine (1–2 eq) (obtained for Procedure A), substituted phenylboronic acid (1–2 eq), $K_2CO_3$

(3 eq), and [1,1′-bis(diphenylphosphino)ferrocene]dichloropalladium(II), complex with dichloromethane (Pd(dppf)Cl2:DCM, 0.1 eq) were dissolved in 1,4-dioxane: $H_2O$ (2:1 mixture) in a microwave tube. Microwave irradiation was carried out at 120 °C for 12 min. After LCMS showed complete conversion to product of interest, the reaction mixture was dissolved in DCM, washed twice with water and once with saturated brine solution. The organic portion was dried with $MgSO_4$, filtered and concentrated, before being purified by column chromatography on SNAP (Biotage) automated purification system using isohexane: ethyl acetate (0–100%).

### Procedure D (Suzuki coupling 2)

This method was used for some compounds for which Procedure C failed to yield the product of interest. 2,6-Dichloropyrazine (1–2 eq), substituted phenylboronic acid (1–2 eq), $K_2CO_3$(aq) or $Na_2CO_3$(aq) (2.0 M, 2–3 eq), and tetrakis(triphenylphosphine)-palladium(0) (Pd(PPh3)4, 0.05 eq) were dissolved in toluene: ethanol (2:1 mixture). The reaction was carried out using either one of the two following procedures:

(i) In a microwave tube, microwave irradiation was carried out at 70 °C for 30 min. After LCMS showed complete conversion to product of interest, the reaction mixture was dissolved in DCM, washed twice with water and once with saturated brine solution. The organic portion was dried with $MgSO_4$, filtered and concentrated, before being purified by column chromatography on SNAP (Biotage) automated purification system using isohexane: ethyl acetate (0–100%).

(ii) In a round-bottom flask, the reaction mixture was refluxed at 120 °C for 7 h. After LCMS showed complete conversion to product of interest, the solvent was removed under vacuum. The crude product was dissolved in DCM, washed twice with water and once with saturated brine solution. The organic portion was dried with $MgSO_4$, filtered and concentrated, before being purified by column chromatography on SNAP (Biotage) automated purification system using isohexane: ethyl acetate (0–100%).

## Analysis of structure–activity relationships in Series 3 and 4

MS_18 and MS_63 were modified to probe for the effects of (i) mono vs. multiple substitutions; (ii) positional isomers; (iii) bioisosteric replacement, e.g., O to S; (iv) electron-withdrawing vs. -donating groups; (v) bulkier vs. smaller groups; and (vi) amino vs. amide linker, to yield compounds in Series 3A, 3B, 4A and 4B.

For 3A analogs, we found that: (i) 4-methoxy and 4-methyl were preferred over the 2- or 3- regio isomeric substituents (MS_18 more potent than Np75-3A01, -3A02, and Np75-3A21 better than -3A19 and -3A20); (ii) when 4-methoxy (4-$OCH_3$) was replaced with the isosteric 4-methylthio (4-$SCH_3$), or 4-methylsulfonyl (4-$SO_2CH_3$) groups, activity was lost; (iii) multiple electron-donating methoxy substitutions, as well as di- and trimethyl groups were not favored; (iv) electron-withdrawing groups such as nitro ($NO_2$), cyano (CN), trifluoromethyl ($CF_3$), carboxylic acid (COOH), and halogen motifs were also unfavorable; (v) bulkier pentyloxy, phenoxy and benzyloxy groups did not also improve activity; (vi) when the amine linker was converted to an amide in Np75-3A49 and Np75-3A51, activity was lost. Np75-3A21 was identified as the best hit from this series.

For 3B analogs: (i) none of the modifications employed improved the activity of MS_18; (ii) the best compound in this series was Np75-3B03, which possessed the same 4-bromo-3-chloro substitution found in Div17E5 but still less potent than MS_18.

For 4A analogs: (i) unlike Series 3A, multiple electron-donating methoxy substitutions (specifically 3,4-dimethoxy), as well as dimethyl groups (i.e., 2,3-dimethyl) were favored in this series; (ii) electron-withdrawing and bulkier phenoxy and benzyloxy motifs were also not preferred in this series. The best compound in the 4A series was Np75-4A22.

For 4B analogs, none of the modifications employed improved the activity of MS_63.

## NMR sample preparation, data acquisition and processing

### Protein expression and purification

TMD fragments of p75[NTR] (residues 240–274), TNFR2 (residues 258–290), and TrkB (residues 419–466) were synthesized by General Biol. The expression constructs were generated by fusing each TMD to the C-terminus of the His9-TrpLE sequence in the pMM-LR6 vector, with a methionine inserted in between for subsequent cleavage during purification, as previously described (Fu et al, 2019). Each protein construct was expressed in transformed E. coli strain BL21 (DE3) cells cultured in LB or M9 minimal media (for isotopic labeling). Cultures were grown at 37 °C until reaching an optical density of ~0.6, then cooled to 25 °C before induction with 0.1 mM isopropyl β-D-thiogalactopyranoside (IPTG). Protein expression was carried out at 25 °C for 18–24 h. After growth, cells were harvested, resuspended in lysis buffer (50 mM Tris, pH 8.0, 200 mM NaCl, 20 mM β-ME), and lysed by sonication. Inclusion bodies were isolated by centrifugation at $25,400 \times g$ and resuspended in denaturing buffer (1% Triton X-100, 6 M guanidine hydrochloride, 50 mM Tris, pH 8.0, 200 mM NaCl, 20 mM β-ME). The inclusion bodies were homogenized using a glass tissue grinder, dissolved, and centrifuged again at $25,400 \times g$. The fusion proteins were bound to nickel affinity resin (Sigma-Aldrich), washed with 8 M urea and deionized water ($dH_2O$), and eluted with 90% formic acid (FA). Each protein was cleaved from TrpLE by hydrolyzing the peptide bond at the C-terminus of the methionine residue with cyanogen bromide (CNBr) (~0.1 g/mL) in 90% FA for 1 h. The reaction mixture was dialyzed (MWCO 1 kDa) to remove excess CNBr and FA, followed by lyophilization. Protein powder was dissolved in 90% FA and purified by reverse-phase high-pressure liquid chromatography (RP-HPLC) using a Zorbax SB-C3 column (Agilent Technologies, Santa Clara, CA) using different gradient buffers. For p75NTR and TrkB TMDs, a gradient from 95% $dH_2O$, 5% acetonitrile (ACN), 0.1% trifluoroacetic acid (TFA) (buffer A) to 100% ACN, 0.1% TFA (buffer B) was applied. The TNFR2 TMD was purified using a gradient from 100% $dH_2O$, 0.1% TFA (buffer A) to 100% methanol, 0.1% TFA (buffer B). Fractions containing pure protein were collected and lyophilized.

### Protein reconstitution in bicelles

Bicelle reconstitution followed a previously described procedure (Piai et al, 2017). The lyophilized protein powder was dissolved in 1,1,1,3,3,3-hexafluoro-2-propanol (HFIP) and mixed with 9 mg of 1,2-dimyristoyl-sn-glycero-3-phosphocholine (DMPC) and 27 mg of 1,2-dihexanoyl-sn-glycero-3-phosphocholine (DHPC) (both from Avanti Polar Lipids, Alabaster). For addition of the compound 4A22, the lyophilized protein powder was dissolved in HFIP and mixed with 9 mg of DMPC and 1 mg of 4A22 (to final compound concentration of ~5 mM). The mixture was dried under a nitrogen stream to form a thin film and lyophilized overnight to remove residual organic solvents. The thin films were dissolved in 500 μL of aqueous buffer (20 mM NaPi, 5 mM DTT, pH 7.0) containing 12 mg of DHPC. The reconstituted proteins were concentrated using a Centricon device (EMD Millipore, Billerica, MA; MWCO 3.5 kDa) to a final volume of ~450 μL. The amounts of DMPC and DHPC were assessed by integrating the DMPC and DHPC terminal methyl peaks in the 1D $^1$H NMR spectrum and adjusted to a [DMPC]/[DHPC] ratio (q) of 0.5. The final NMR samples contained 0.3 mM of TMDs, 30 mM DMPC, and 10% (v/v) D2O, in the absence or presence of 5 mM 4A22.

### NMR data acquisition and processing

Two-dimensional $^1$H-$^{15}$N TROSY-HSQC spectra were recorded on a Bruker 600 MHz spectrometer equipped with a cryogenic probe. All measurements were performed at 303 K. NMR data sets were processed using nmrPipe (Delaglio et al, 1995), and the resulting spectra were analyzed using Ccpnmr (Vranken, 2005). Sequence-specific assignment of the p75NTR TMD backbone chemical shifts was accomplished using a set of TROSY-enhanced triple resonance experiments including HNCA, HN(CO)CA, and HNCO (Kay et al, 1990; Salzmann et al, 1999), as well as an ultra-high-resolution 3D $^{15}$N- edited NOESY-TROSY-HSQC (τmix = 200 ms) spectrum as described (Piai et al, 2020).

## Pharmacokinetics and toxicology

Plasma pharmacokinetics of Np75-4A22 was investigated following single intravenous (3 mg/kg) or oral (10 mg/kg) dose administrations in C57BL/6 mice of both sexes. Blood samples were collected from the retro orbital plexus at pre-dose, 0.08, 0.25, 0.5, 1, 2, 5, 10 and 24 hs (for intravenous) and pre-dose, 0.25, 0.5, 1, 2, 4, 5, 10 and 24 hs (for oral). Toxicology of Np75-4A22 was tested in a 3 week repeated dose study in C57BL/6 mice after 200 mg/kg/day oral gavage administration. Pharmacokinetics and toxicology studies were carried out by Sai Life Sciences Limited, Hyderabad, India.

## Melanoma lung invasion assay, immunohistochemistry and immunofluorescence

Eight-week-old female SCID mice (Charles River Laboratories, Shrewsbury, MA, USA) were housed in groups of three to five and maintained on a 12-h light/dark schedule with a temperature of 22 °C ± 1 °C and a relative humidity of 50% ± 5%. Food and water were available ad libitum. This research study was reviewed and approved by the University of Calgary Animal Care Committee (AC22-0054) and McGill University (AUP-JGH-8269) and adheres to established ethical principles and guidelines. A875-GFPLuc (A875-NT) or A875-GFPLuc shp75 (A875-shp75) cells harvested using Puck's EDTA were resuspended in PBS and injected intravenously into SCID mice ($5 \times 10^5$ cells/200 μl per mouse). Np75-4A22 was dissolved in 0.5% sodium carboxymethyl cellulose (NaCMC) in reverse osmosis H$_2$O and administered by oral gavage at 200 mg/kg/day starting one day prior to intravenous (tail vein) injection of tumor cells and continued for 3 1-week cycles (cycle: 5 days on drug, 2 days off drug). Tumor spread to the lung was monitored weekly by luciferase activity using the Xenogen IVIS-200 Optical in vivo luciferase imaging system as described previously (Lun et al, 2016) and by immunohistochemistry (IHC) at designated time points. IHC was performed after formalin fixation, paraffin embedding and sectioning of the lungs, using a human-specific mouse polyclonal nucleolin antibody (1:500 dilution) and detected using Dako liquid DAB+ Substrate Chromogen System (Dako: k3468, California, USA) followed by a hematoxylin counterstain (Sigma, Oakville, ON, Canada). Sections were mounted and imaged on an Olympus phase contrast microscope (Olympus Life Science, Tokyo, Japan). For immunofluorescence, sections were stained with anti-human nucleolin and anti-human p75NTR antibodies (1:1000 dilution), subsequently detected with 1:200 dilutions of Alexa Fluor™ 488 Goat anti-Mouse IgG (H + L) Highly Cross-Adsorbed Secondary Antibody and Alexa Fluor™ 568 Goat anti-Rabbit IgG (H + L) Cross-Adsorbed Secondary Antibody. Sections were further stained with DAPI to detect all nuclei. Between each treatment, TBS Automation Wash Buffer (Biocare Medical TWB945M) was employed. Images were taken with LSM800 confocal microscope and ZEN system.

---

**The paper explained**

**Problem**
Melanoma is the most serious form of skin cancer and among the most common cancers in the Caucasian population. Immune checkpoint inhibitors (ICI) are currently a primary treatment option available to patients with metastatic melanoma, and while these treatments provide long-term survival in some patients, recent clinical trials indicate that 40–50% of metastatic melanoma patients fail to respond to ICI therapy.

**Results**
This study tested the notion that small molecules capable of interfering with the function of death receptor p75NTR, implicated in the metastasis of a wide range of melanomas, may be able to prevent the spread and migration of melanoma cells. We describe the identification of a novel small molecule targeting the transmembrane domain of p75NTR termed Np75-4A22 that is capable of inhibiting migration and lung invasion of human melanoma cells in mice.

**Impact**
These results provide new evidence supporting the feasibility of targeting the transmembrane domains of cell surface receptors for the identification of small molecules capable of modifying receptor activity in useful ways. Np75-4A22 represents a promising lead in the development of therapies against melanoma metastasis which could prove particularly beneficial for patients showing poor responsiveness to conventional chemotherapy or ICI.

---

## Statistical analysis

Statistical Analysis Software (SAS Institute, Inc.) and GraphPad Prism (versions 4 or 8; GraphPad Software Inc, San Diego, CA, USA) were used for statistical analyses. Survival curves were generated using the Kaplan–Meier method. Experimental data was collected from multiple experiments and reported as the treatment mean ± SEM. Statistical significance was calculated using the one-way ANOVA followed by Tukey's multiple comparisons test for all data except that shown in Fig. 8B which used the Student $t$ test. $P$ value of less than 0.05 was considered statistically significant. No blinding was employed in the study. SCID mice were randomly assigned to experimental groups. Sample sizes are indicated in the figure legends. No data points were excluded from the analyses.

## Data availability

This study includes no data deposited in external repositories.

The source data of this paper are collected in the following database record: biostudies:S-SCDT-10_1038-S44321-025-00297-1.

## Peer review information

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

## Acknowledgements

We thank Ket Yin Goh and Annika Andersson for technical assistance. This research was funded by grants to CFI from the Singapore National Medical Research Council (NMRC/OFIRG/MOH-000224), Swedish Research Council (2020-01923), Swedish Cancer Society (18-0670 and 18-0829), Chinese Institute for Brain Research (CIBR, start-up grant) and Peking University (start-up grant); to DLS from the Canadian Cancer Society and Robert C Westbury Endowment; and to JJC from Shanghai Science and Technology Committee Fund (22JC1410500) and National Natural Science Fund of China, Research Fund for International Scientists (82350710799).

## Author contributions

**Vanessa Lopes-Rodrigues**: Formal analysis; Investigation; Methodology; VL-R performed the majority of cell culture experiments, screening and validation of all Div17E5 derivatives and compiled a preliminary draft of the manuscript and figures. **Samuel A Nyantakyi**: Investigation; Methodology; SAN performed all medicinal chemistry and SAR studies resulting in compounds of Series 3 and 4 including the lead compound Np75-4A22. **Xueqing Lun**: Investigation; Methodology; XL performed the lung invasion studies in SCID mice. **Xueyan Han**: Investigation; Methodology; XH performed the phalloidin assay of filopodia formation and the Boyden chamber assay in WM115 cells. **Jianbo Zhang**: Investigation; Methodology; JZ performed the lung invasion studies in SCID mice. **Ajeena Ramanujan**: Investigation; Methodology; AR performed co-

immunoprecipitation studies. **Shuhailah Salim**: Investigation; Methodology; performed the initial screening that identified Div17E5. **Michael Saleeb**: Investigation; Methodology; performed preliminary medicinal chemistry studies resulting in compounds of series 1 and 2. **Liane Babes**: Investigation; Methodology; performed additional cell migration studies. **Angela Z Chou**: Investigation; Methodology; developed bicelle NMR systems and recorded NMR spectra. **Lingyu Du**: Investigation; Methodology; developed bicelle NMR systems and recorded NMR spectra. **Siyi Dong**: Investigation; Methodology; developed bicelle NMR systems and recorded NMR spectra. **James J Chou**: Supervision; Funding acquisition; supervised the NMR work. **Donna L Senger**: Conceptualization; Supervision; Funding acquisition; designed and supervised all cancer studies. **Carlos F Ibáñez**: Conceptualization; Supervision; Funding acquisition; Writing—original draft; Project administration; Writing—review and editing; conceived the original project, contributed to project supervision, coordination and data analysis, and wrote the manuscript.

Source data underlying figure panels in this paper may have individual authorship assigned. Where available, figure panel/source data authorship is listed in the following database record: biostudies:S-SCDT-10_1038-S44321-025-00297-1.

## Funding

## Disclosure and competing interests statement

The authors declare no competing interests.

# Expanded View Figures

Series 1/2= 41 compounds

**Figure EV1.** Chemical structures of Div17E5 analogs: Series 1 and 2.

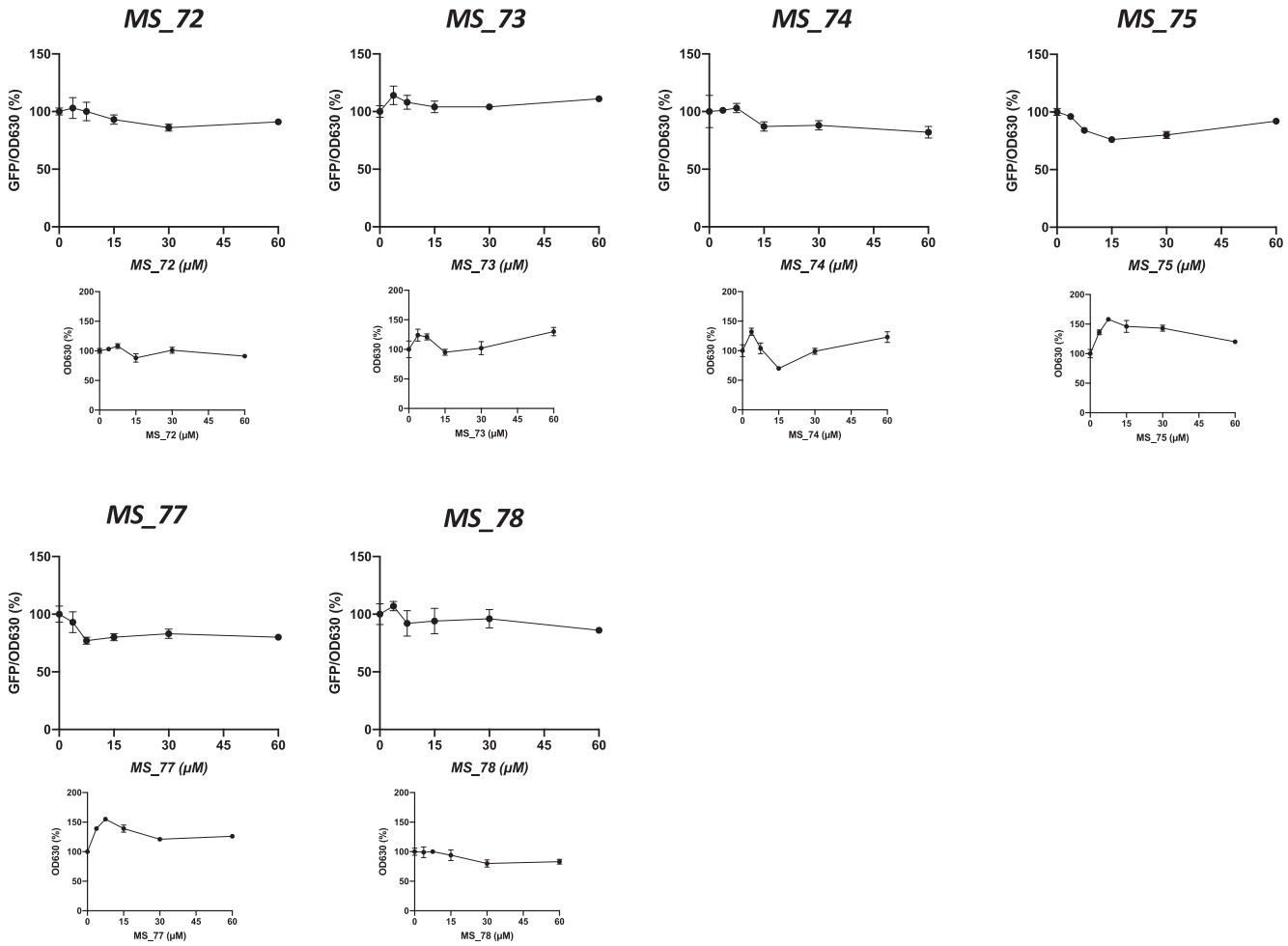

**Figure EV2.** GFP/OD630 and OD630 of compounds in Series 2A.

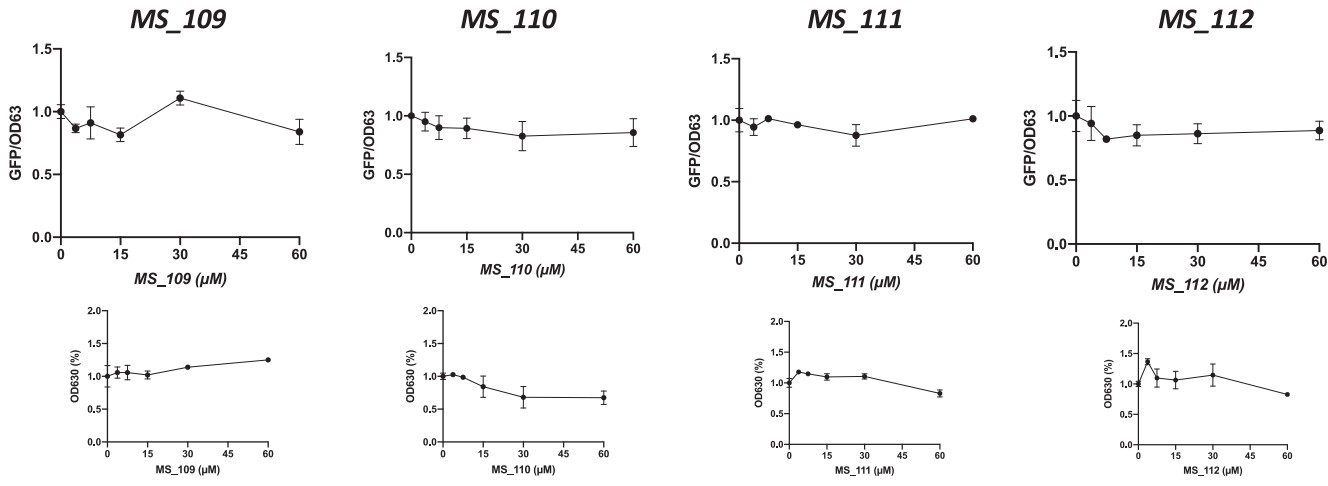

**Figure EV3.** GFP/OD630 and OD630 of compounds in Series 2B.

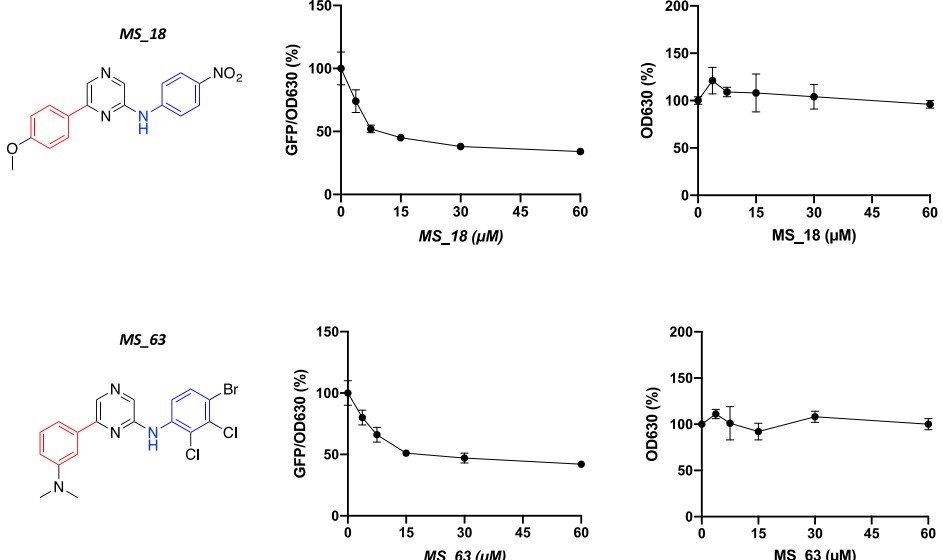

**Figure EV4.** GFP/OD630 and OD630 of compounds MS_18 and MS_63.

3A Series = 51 compounds

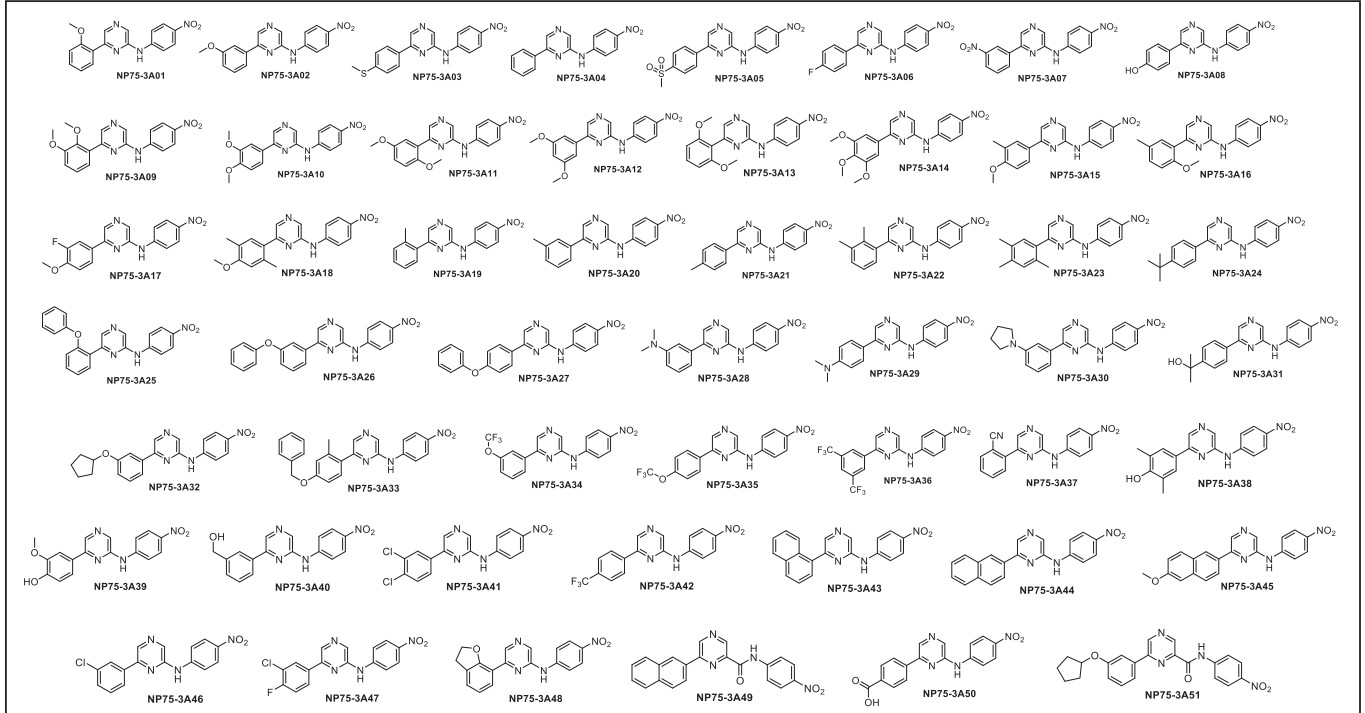

**Figure EV5.** Chemical structures of Div17E5 analogs: Series 3A.

3B Series = 41 compounds

**Figure EV6.** Chemical structures of Div17E5 analogs: Series 3B.

4A Series = 27 compounds

**Figure EV7.** Chemical structures of Div17E5 analogs: Series 4A.

4B Series = 25 compounds

**Figure EV8.** Chemical structures of Div17E5 analogs: Series 4B.

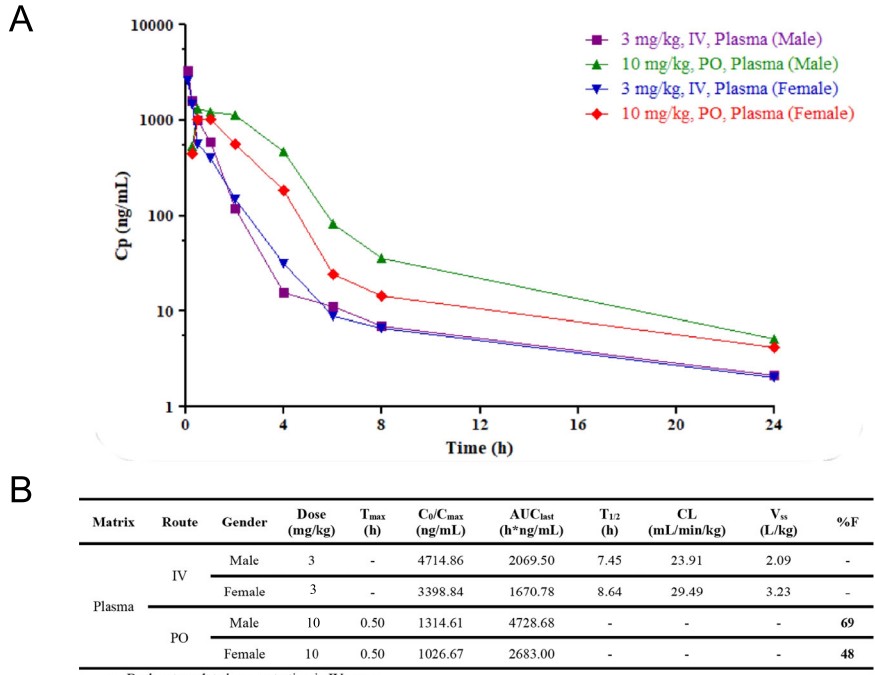

**A**

**B**

| Matrix | Route | Gender | Dose (mg/kg) | $T_{max}$ (h) | $C_0/C_{max}$ (ng/mL) | $AUC_{last}$ (h*ng/mL) | $T_{1/2}$ (h) | CL (mL/min/kg) | $V_{ss}$ (L/kg) | %F |
|---|---|---|---|---|---|---|---|---|---|---|
| Plasma | IV | Male | 3 | - | 4714.86 | 2069.50 | 7.45 | 23.91 | 2.09 | - |
| | | Female | 3 | - | 3398.84 | 1670.78 | 8.64 | 29.49 | 3.23 | – |
| | PO | Male | 10 | 0.50 | 1314.61 | 4728.68 | - | - | - | 69 |
| | | Female | 10 | 0.50 | 1026.67 | 2683.00 | - | - | - | 48 |

a – Back extrapolated concentration in IV group.

**Figure EV9. Pharmacokinetic study of Np75-4A22 in mice.**

(**A**) Graph shows plasma concentrations of Np75-4A22 in male and female C57BL/6 mice ($N = 9$) after a single intravenous (3 mg/kg) or oral (10 mg/kg) administration of 10 mg/kg. Plasma was collected at pre-dose, 0.25, 0.5, 1, 2, 4, 5, 10 and 24 hs and assessed for Np75-4A22 using LC/MS/MS. (**B**) Pharmacokinetic parameters of Np75-4A22 in plasma following a single intravenous (3 mg/kg) or oral (10 mg/kg) in male and female C57BL/6 mice. Cmax: maximum (peak) plasma drug concentration (amount/volume). Tmax: time to reach maximum (peak) plasma concentration following drug administration. AUClast: area under the plasma concentration-time curve from time zero to time of last measurable concentration (time/volume). Bioavailability (%F): extent and rate at which the active moiety (drug or metabolite) enters systemic circulation.

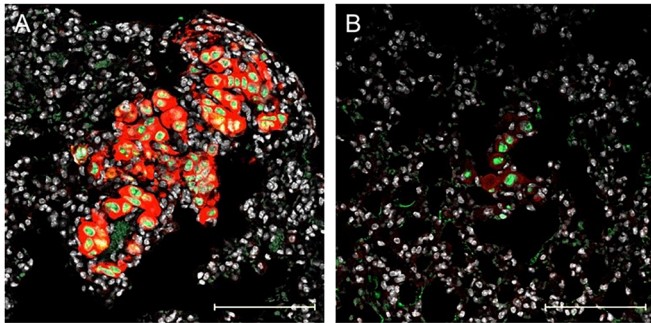

**Figure EV10.  Histological analysis of p75NTR expression in lung metastasis produced by A875 cells.**

p75NTR immunostaining (red) in lung metastasis induced by A875-NT (**A**) or shp75-A875 (**B**) cells counter-stained for human nucleolin (green) and DAPI (white). Scale bar, 100 µM.

