## [Peer Review File · EMBO Molecular Medicine]

Impaired migration and lung invasion of human melanoma by a novel small molecule targeting the ...

Vanessa Lopes-Rodrigues, Samuel Nyantakyi, Xueqing Lun, Xueyan Han, Jianbo Zhang, Ajeena Ramanujan, Shuhailah Salim, Michael Saleeb, Liane Babes, Angela Chou, Lingyu Du, Siyi Dong, James Chou, Donna Senger, and Carlos Ibanez

Corresponding author: Carlos Ibanez (carlos.ibanez@pku.edu.cn)

Review Timeline:

Submission Date:	11th Dec 23
Editorial Decision:	4th Jan 24
Revision Received:	10th May 25
Editorial Decision:	6th Jun 25
Revision Received:	18th Jun 25
Editorial Decision:	1st Jul 25
Revision Received:	21st Jul 25
Editorial Decision:	1st Aug 25
Revision Received:	11th Aug 25
Accepted:	11th Aug 25

Editor: Lise Roth

Transaction Report:

4th Jan 2024

Dear Prof. Ibanez,

Thank you for the submission of your manuscript to EMBO Molecular Medicine. We have now heard back from the referees who agreed to evaluate your manuscript. As you will see below, the reviewers raise substantial concerns on your work, which unfortunately preclude its publication in EMBO Molecular Medicine in its current form.

The reviewers find that the question addressed by the study is of potential interest, however, they remain unconvinced that some of the major conclusions are sufficiently supported by the data. They thus raise the following major issues:

- define the structural basis for the differential effects of Div175 and Np75-4A22
- validation in a conventional metastasis model
- validation in additional cellular models

If you feel you can satisfactorily address these points and those listed by the referees, you may wish to submit a revised version of your manuscript. Please attach a covering letter giving details of the way in which you have handled each of the points raised by the referees. A revised manuscript will once again be subject to review, and we cannot guarantee at this stage that the eventual outcome will be favorable.

We are expecting your revised manuscript within three months, if you anticipate any delay, please contact us.

We require:

4) A .docx formatted letter INCLUDING the reviewers' reports and your detailed point-by-point responses to their comments. As part of the EMBO Press transparent editorial process, the point-by-point response is part of the Review Process File (RPF), which will be published alongside your paper.

5) A complete author checklist, which you can download from our author guidelines (<https://www.embopress.org/page/journal/17574684/authorguide#submissionofrevisions>). Please insert information in the checklist that is also reflected in the manuscript. The completed author checklist will also be part of the RPF.

6) It is mandatory to include a 'Data Availability' section after the Materials and Methods. Before submitting your revision, primary datasets produced in this study need to be deposited in an appropriate public database, and the accession numbers and database listed under 'Data Availability'. Please remember to provide a reviewer password if the datasets are not yet public (see <https://www.embopress.org/page/journal/17574684/authorguide#dataavailability>).

7) For data quantification: please specify the name of the statistical test used to generate error bars and P values, the number (n) of independent experiments (specify technical or biological replicates) underlying each data point and the test used to calculate p-values in each figure legend. The figure legends should contain a basic description of n, P and the test applied. Graphs must include a description of the bars and the error bars (s.d., s.e.m.). Please provide exact p values.

8) Our journal encourages inclusion of *data citations in the reference list* to directly cite datasets that were re-used and obtained from public databases. Data citations in the article text are distinct from normal bibliographical citations and should

directly link to the database records from which the data can be accessed. In the main text, data citations are formatted as follows: "Data ref: Smith et al, 2001" or "Data ref: NCBI Sequence Read Archive PRJNA342805, 2017". In the Reference list, data citations must be labeled with "[DATASET]". A data reference must provide the database name, accession number/identifiers and a resolvable link to the landing page from which the data can be accessed at the end of the reference. Further instructions are available at .

9) We replaced Supplementary Information with Expanded View (EV) Figures and Tables that are collapsible/expandable online. A maximum of 5 EV Figures can be typeset. EV Figures should be cited as 'Figure EV1, Figure EV2" etc... in the text and their respective legends should be included in the main text after the legends of regular figures.

10) The paper explained: EMBO Molecular Medicine articles are accompanied by a summary of the articles to emphasize the major findings in the paper and their medical implications for the non-specialist reader. Please provide a draft summary of your article highlighting

11) For more information: There is space at the end of each article to list relevant web links for further consultation by our readers. Could you identify some relevant ones and provide such information as well? Some examples are patient associations, relevant databases, OMIM/proteins/genes links, author's websites, etc...

12) Author contributions: CRediT has replaced the traditional author contributions section because it offers a systematic machine readable author contributions format that allows for more effective research assessment. Please remove the Authors Contributions from the manuscript and use the free text boxes beneath each contributing author's name in our system to add specific details on the author's contribution. More information is available in our guide to authors.

13) Disclosure statement and competing interests: We updated our journal's competing interests policy in January 2022 and request authors to consider both actual and perceived competing interests. Please review the policy <https://www.embopress.org/competing-interests> and update your competing interests if necessary.

14) Every published paper now includes a 'Synopsis' to further enhance discoverability. Synopses are displayed on the journal webpage and are freely accessible to all readers. They include a short stand first (maximum of 300 characters, including space) as well as 2-5 one-sentences bullet points that summarizes the paper. Please write the bullet points to summarize the key NEW findings. They should be designed to be complementary to the abstract - i.e. not repeat the same text. We encourage inclusion of key acronyms and quantitative information (maximum of 30 words / bullet point). Please use the passive voice. Please attach these in a separate file or send them by email, we will incorporate them accordingly.

15) As part of the EMBO Publications transparent editorial process initiative (see our Editorial at <http://embomolmed.embopress.org/content/2/9/329>), EMBO Molecular Medicine will publish online a Review Process File (RPF) to accompany accepted manuscripts.

In the event of acceptance, this file will be published in conjunction with your paper and will include the anonymous referee reports, your point-by-point response and all pertinent correspondence relating to the manuscript. Let us know whether you agree with the publication of the RPF and as here, if you want to remove or not any figures from it prior to publication. Please note that the Authors checklist will be published at the end of the RPF.

I look forward to receiving your revised manuscript.

Yours sincerely,

Lise Roth

***** Reviewer's comments *****

Referee #1 (Remarks for Author):

This manuscript by Lopes-Rodriguez and colleagues builds upon prior studies by the Ibanez lab, which had previously identified a compound (NSC49652, using an Ara TM screen) which upon binding, alters the conformation of the transmembrane domain (TM) of p75, and affects melanoma cell viability (Goh et al, Cell Chemical Biology, 2020). In the current manuscript, this approach is extended using a larger library, and a positive hit was subject to further chemical elaboration to identify more potent compound.

In the 2020 paper, compound NSC49652, was shown to bind to the N-terminal region of the TM domain of p75, deduced using NMR, and induce apoptosis of A875 melanoma cells, which express p75 at high levels. The initial positive hit described here, Div17E5, bears some structural similarities to NSC49652 and also induces A875 apoptosis. Mutagenesis of two N-terminal amino acids near the N-terminal region of the TM domain impairs this effect, suggesting that Div175 may bind in a similar region to induce a conformational shift in the interaction of p75 TM domains. Div175 also impaired cell migration in a p75-dependent manner. However, a more potent analog, Np75-4A22, only impairs cell migration, and using mutagenesis of the TM domain, interacts with only one of the two aa previously identified. These results suggest that these compounds interact differentially with the TM domain, and induce distinct activities on downstream signaling by p75.

These results provide an important opportunity to evaluate structure: function analysis of how conformational change in the TM domain of p75 may regulate the specificity of downstream signaling, a critical question for the field. The authors need to determine the structural basis for these differences- do the 2 compounds bind to different regions, and differentially alter TM:TM interactions?

Lastly, the in vivo effects of Np75-4A22 need to be dissected more comprehensively, specifically to evaluate the distinct activities on tumor growth and migration, as these are important preclinical studies. Unlike the 2020 Goh et al paper, in which A875 melanoma cells (or A875-shp75 knockdown cells) were injected subcutaneously into the flank to establish a primary tumor which then metastasized to lungs (the typical model), in the current study the authors inject cells IV (assume tail vein, but not specified) and assess effects on growth in multiple organs (by whole animal imaging) or in lung tissue sections. This is NOT a model that evaluates metastasis as no primary tumor is established. It is a model which evaluates the ability of cells to become established and grow locally in organs following intravascular injection, which disseminates tumor cells throughout the body. The authors should perform a true xenograft model (as they did in Goh et al), which will provide important data on growth of cells in a primary site (augmenting in vitro studies, to bolster the conclusion that Np75-4A22 does not affect cell growth/apoptosis), and also assesses metastasis to lung or other organs. From the data provided, one cannot conclude that the most prominent effect of the drug is on migration/metastasis.

1. A comparison of the biological effects of Div175 and the more potent analog Np75-4A22 provides an important opportunity for structure: function analysis which should be explored. Using A875 cells, Div175 both induces apoptosis and impairs cells migration, whereas Np75-4A22 does not induce apoptosis, but does impair migration. The authors need to perform NMR to document where each of these compounds binds to the TM of p75, and determine how each perturbs TM:TM dimerization.
2. The data regarding the effects of Div175 in impairing NGF-induced migration in the Boyden chamber assay is not provided (legend for Fig 3D, but no data is provided with A875shp75 cells).
3. The description of results relating to Fig 7A does not align with the data. The interaction of fascin with p75, and its enhanced interaction with NGF treatment was previously described and it is stated that this is the case here. Yet the data provided indicates that NGF reduces the interaction of fascin and p75. Please clarify.

4. Fig 8. A xenograft model (injection of cells into the flank to establish a primary tumor, followed by whole animal imaging to assess primary tumor growth and growth of metastases) should be undertaken. In addition, for the IV injection model, whole animal images from all 4 conditions should be provided (A875 scrambled sh +/- Np75-4A22) and (A875-shp75 +/-Np75-4A22) should be shown.

5. Clarifying language.

a. The authors need to quote the literature more broadly, and review terminology. For example, on page 4, rather than state that "The mechanism by which p75 becomes activated" this should be restated as "A mechanism by which..." and also include a description of work by other labs (Marchetti, et al PNAS 2019 doi: 10.1073/pnas.1902780116) indicating that fast-diffusing p75 monomers induce apoptosis and growth cone collapse by neurotrophins.

b. Also on page 4, the authors imply that the Cys-mutant in the TM of p75 impairs p75 action in an AD mouse model- but in fact, this work suggests that what is affected is trafficking of Abeta, not p75 signaling.

c. On page 9, the authors refer to a melanoma invasion model, but this is simply an in vitro migration model- this should be corrected in Fig 3 and in Fig 6 and in the text that refers to both figures.

d. The authors use A875, A875-WT, A875-NT seemingly interchangeably. Is the A875-NT a real control (ie scrambled shRNA) or is it just wildtype cells.

Referee #2 (Comments on Novelty/Model System for Author):

The authors used one melanoma line in vitro and in vivo to test a new compound. It is adequate in the first stage of drug discovery. Though it would be a more solid argument if they had used more models to validate their findings.

Referee #2 (Remarks for Author):

The p75NT has been associated with tumor metastases. The dimerization of p75NT transmembrane domain (TMD) transfers extracellular signals to intracellular regulation of apoptotic cell death and cell migration. TMD of p75NT has a unique sequence that could be a potential target. The authors applied AraTM assay to screen over 8,000 compounds and found Div17E5 could disturb TMD-mediated dimerization. They showed that Div17E5 induced apoptosis and inhibit cell migration in A875 cell line. Using medicinal chemistry, a derivative termed Np75-4A22 was identified for lower IC50 to inhibit cell migration without inducing apoptosis. Np75-4A22 abolished the binding of fascin with p75NT and reduce fascin recruitment into the detergent insoluble fraction of A875. They tested regimens for Np75-4A22 in A875 mouse model and found significant inhibition of melanoma lung metastases. The function and mechanism of the novel compound are appealing that it could become a new therapeutic strategy for melanoma metastasis. However, there are issues that need to be addressed or discussed further.

In Figure 3B, wound healing assay showed that A875 shp75 cells is as invasive as A875 NT cells that both wound areas decreased from 1 to ~0.75 in 24 hours. But in Figure 8C, A875 shp75 showed significant less metastasis in the lungs in SCID mice. Do the in vitro assays recapitulate melanoma invasiveness? If not, is it possible the ideal compounds were missed from the screening in melanoma cells (Figure 6)?

In Figure 7A, co-IP experiment requires input and IgG control to show positivity and specificity. In Figure 7B, quantification using the ratio of Triton-insoluble fascin to total fascin will be much more solid than only showing the western blot.

Also in Figure 7A, the labelling of NGF +/- does not match what is described in the text "NGF treatment strongly stimulated the association of fascin with p75NTR".

Referee #3 (Remarks for Author):

In this manuscript, Lopes-Rodrigues and colleagues identified a novel small molecule inhibiting p75NTR-mediated migration of human melanoma cells. Although significant progress has been made in the treatment of melanoma with BRAF inhibitors or immune checkpoint inhibitors, about 40% of patients do not respond to these medicines. p75NTR is one of the death receptor family present in melanoma cells and is known to induce apoptotic cell death or regulate cell motility and migration. They used a transmembrane domain (TMD) of p75NTR as a drug target and identified Div17E5 from a chemical library of about 8,000 compounds using the AraTM assay-based screening. Furthermore, they applied medical chemistry and identified a more potent derivative, Np75-4A22, that blocks NGF-mediated melanoma invasion. The authors provided evidence with in vitro assays to show the function of Div17E5 and Np75-4A22 in apoptotic cell death and impairing cell motility/chemotaxis of the melanoma cell line. In addition to in vitro analysis, they studied the oral administration effect of Np75-4A22 in the mouse xenograft model and provided evidence that Np75-4A22 reduced the spread and seeding of melanoma cells.

Overall, the study is well executed and interesting to the readership of EMBO Molecular Medicine. The study provides new evidence of the feasibility of identifying small chemical compounds targeting the TMDs of transmembrane receptors and modifying their activity, as well as an alternative/complementary approach for patients who do not respond to currently available therapy for melanoma. Biologically, it is interesting that Div17E5 and Np75-4A22 showed distinct functional downstream outcomes (anti-apoptotic vs motility), though they are targeting the same p75NTR TMD.

There are several points and weakness that should be addressed prior to publication.

Major points:

1. The identification of a small molecule Np75-4A22 and finding its function to inhibit p75NTR-mediated migration/metastasis of melanoma cells is the most interesting point of the manuscript. The authors showed the difference between newly identified Np75-4A22 and another chemical substrate, Div17E5, which also affects p75NTR signaling in terms of their function to induce apoptotic cell death and cell motility. A series of in vitro experiments indicated that the Np75-4A22 did not induce apoptosis of melanoma cells while Div17E5 did, but was more potent at blocking melanoma cell motility. Figure 8 shows an in vivo study demonstrating that Np75-4A22 reduced A875 melanoma cells using a mouse xenograft model. Although the in vitro studies showed solid evidence that Np75-4A22 has more potent to block cell motility compared to Div17E5, it is not clear about its advantage in in vivo study over Div17E5. The data shown in Figure 8 indicated that it reduced the tumor growth of melanoma cells. However, the same result probably could be obtained from the same experiments using pro-apoptotic chemicals. To support the idea that the activity of blocking cell motility of Np75-4A22 would be superior to Div17E5, I believe some additional data are needed. e.g., Showing the same type of in vivo data with Div17E5 and seeing whether Np75-4A22 is more effective. Also, in Figure 8C, the authors compared the number of cancer cells among the conditions, but if they emphasized the anti-migration/metastasis function of Np75-4A22, it would be interesting whether there is a difference in the number of cancer lesions between these conditions. If the number of lesions in the condition with Np75-4A22 is less than that in Div17E5, it would be another evidence to concrete their insistence.
2. The authors think that one of the mechanisms that Np75-4A22 impairs NGF-mediated melanoma migration is via reducing fascin/p75NTR interaction. Because fascin is an actin-bundling protein, they should examine if Np75-4A22 treatment cause the structural changes of actin cytoskeleton such as lamellipodia formation or filopodia formation. The labeling in figure 7A seems to be wrong. Please check and correct it.
3. One weakness of this manuscript is the unclearness of the statistical analyses. The method section mentions that the data were shown as mean{plus minus}SEM, and statistical significance was calculated using the Student t-test, Mann Whitney, or one-way ANOVA. However, in many of the figures with statistical analyses, there is no description of which statistical test was used. The methods should be described in each legend. Also, if they used the Mann-Whitney test, I assume some of the data distribution was non-parametric. If so, they should use other ways to display the data, such as the box-and-whisker plot. Using mean{plus minus}SEM for non-parametric data often causes misleading.
4. Another weakness is the biological specificity of using p75NTR as a drug target. It has been well known that p75NTR is expressed in many normal cell types through various organs. They mention, 'No mortality or adverse clinical signs, including body weight, food consumption or effects on gross histological pathology assessed on formalin fixed paraffin embedded lung, liver, kidney or brain were observed in any of the Np75-4A22 treated mice. However, in the brain, it has also been known that p75NTR has an important role in memory formation, which regular histological pathology cannot assess. Although I don't necessarily think they should add such data, it would be better to discuss it as a potential side effect of the compound.
5. The authors used the AraTM transcription factor system in bacteria for chemical compound screening and identified 4 potential compounds (Figure 1A). However, it was unclear how they further narrowed it down to the one (Div17E5). In the text, they mention 'and subjected to secondary and tertiary screens for reduced toxicity and increased potency and specificity.' I could not figure out what assays were used for these additional screenings. During the screening procedure, it is very important to narrow down to the small number from many candidates. It would be useful information for the readers on how they perform the secondary and tertiary screens.

Minor points:

Figure 1:

Panel names in the main text and the actual panels are not consistent. e.g. p6. I8 and I9, Figure 1D, E should be 1D, 1F, and p6. I9, 1F should be 1E.

Figure notes in both 1G and 1H use colored squares, but the figures use a square and a circle.

In the text, they say, 'The compound induced oscillations in real-time anisotropy (Figure 1G)'; however, the figure seems that the vehicle and Div17E5 were flipped.

Figure 2:

The order of figure legends is inconsistent with the text and figures. Please correct them.

Figure 3:

I believe the lower panel of current B should be C. C should be D, and D should be E.

In the figure legend for (A), it is not clear what 'Size bar, Xmm' means.

It is probably better to add references for p7. I13 ('As p75NTR is known to regulate motility and migration in melanoma cells').

Figure 4:

p8. I20, the authors claim, 'However, when tested in the wound healing assay in A875-NT cells, both compounds were less potent than Div17E5.' There is no statistical analysis on this panel (4D), so the overstatement should be avoided. If they want to clearly state 'were less potent,' then the statistical analysis, such as two-way ANOVA, should be added.

Figure 5:

Panel B has no asterisks for the stats.

Figure 6:

Figure 6A, as the conditions are defined quantitatively, it would be more informative if they could present quantitative analysis for the WB signals (this can also be possible for Figure 2B). Since it is described that the experiment was repeated three times with comparable results, this is feasible.

The note for Figure 6B (bottom) misses symbols for A875 shp75 + drug and A875 shp75 + vehicle.

Responses to reviewers' comments (EMM-2023-19128-T)

Reviewer #1

1. ***A comparison of the biological effects of Div175 and the more potent analog Np75-4A22 provides an important opportunity for structure:function analysis which should be explored. Using A875 cells, Div175 both induces apoptosis and impairs cells migration, whereas Np75-4A22 does not induce apoptosis, but does impair migration. The authors need to perform NMR to document where each of these compounds binds to the TM of p75, and determine how each perturbs TM:TM dimerization.***

Due to the amount of effort required for 3D structure determination of drug:TMD complexes by multidimensional NMR, and upon consultation with the editor, it was agreed that such NMR studies would be "beyond the scope of the current study". Nevertheless, we present in the revised version data from 2D NMR with cross-peak identification for the interaction of Np75-4A22 with three different TMDs, namely those of p75NTR, TrkB and TNFR2 (Figure 5F-H). As explained in the revised manuscript, significant chemical shifts were detected in a number of residues in the TMD of p75NTR upon addition of the compound. Interestingly, assignment of the cross-peaks validated Ile252 as one of the key residues undergoing conformational change upon Np75-4A22 binding, in agreement with the mutagenesis data. Importantly, no or minimal chemical shift perturbations were observed upon addition of the compound to the two unrelated TMDs. These results demonstrate the specific interaction of the compound with the p75NTR TMD and identify several TMD residues that are either directly involved in drug binding or significantly affected by it.

2. ***The data regarding the effects of Div175 in impairing NGF-induced migration in the Boyden chamber assay is not provided (legend for Fig 3D, but no data is provided with A875shp75 cells).***

This figure legend incorrectly referred to shp75-A875 cells while the data presented were only on NT-A875 (control) cells. This figure legend has been amended to correct this error. We note that shp75-A875 cells do not respond to NGF (as shown in panel E of the same figure), explaining why the assay was not performed in those cells.

3. ***The description of results relating to Fig 7A does not align with the data. The interaction of fascin with p75, and its enhanced interaction with NGF treatment was previously described and it is stated that this is the case here. Yet the data provided indicates that NGF reduces the interaction of fascin and p75. Please clarify.***

The labels in the Western blot shown in this panel were incorrect. They have now been amended.

4. ***Fig 8. A xenograft model (injection of cells into the flank to establish a primary tumor, followed by whole animal imaging to assess primary tumor growth and growth of metastases) should be undertaken. In addition, for the IV injection model, whole animal images from all 4 conditions should be provided (A875 scrambled sh +/- Np75-4A22) and (A875-shp75 +/-Np75-4A22) should be shown.***

This request to study metastasis after injection of cells into the flank of mice to establish a primary tumor, rather than i.v. injection, would seem obvious to most people, except those that actually work in the field of tumor metastasis. The truth is that this is much easier said than done, as there are no good models that spontaneously metastasize from a primary tumor before the primary tumor is too large and the animal needs to be sacrificed, which is why researchers in the metastasis field inject tumors cells intravenously. During the past year, we performed numerous attempts on this approach, varying the times of implantation, drug administration and lung analysis in different ways. These studies were very involved and time-consuming, which explains the long delay in completing the revisions to the

manuscript. Long story short, it was really difficult to get sufficient numbers of cells metastasized to the lung before the flank tumors reached a size that required sacrificing the animals. Although a few metastatic cells did appear in the lungs at the 5-week time point, numbers were too low to perform a meaningful quantitative (and statistical) assessment of the effects of the drug. Frustrating outcome after investing more than a year and so much effort, which in a way reflects the difficulties that this field as a whole faces to perform robust melanoma metastasis studies in mice.

After consultation with the editor, it was agreed to omit this approach and instead i) “*tone down the claims about metastasis, emphasizing instead that the compound affects cell invasion and motility*”, and ii) present data showing that Np75-4A22 can inhibit cell migration in another melanoma cell line. We have done this in the revised version of the manuscript. We no longer refer to the effects of the compounds as affecting melanoma metastasis, but instead cell migration, motility or invasion as appropriate. This is also reflected in the revised title. We also present evidence in the Boyden chamber assay that Np75-4A22 can inhibit NGF-induced cell migration of the WM115 human melanoma cell line (Figure 6E), which also expresses p75^{NTR}.

Whole animal images from all 4 conditions are now provided.

5. **Clarifying language.**

a. The authors need to quote the literature more broadly, and review terminology. For example, on page 4, rather than state that “The mechanism by which p75 becomes activated” this should be restated as “A mechanism by which...” and also include a description of work by other labs (Marchetti, et al PNAS 2019 doi: 10.1073/pnas.1902780116) indicating that fast-diffusing p75 monomers induce apoptosis and growth cone collapse by neurotrophins.

We agree to the wording change proposed by reviewer but respectfully decline to include the reference requested. While we obviously have no knowledge of a possible involvement of the reviewer in the paper mentioned, and with all due respect to the reviewer, I (C.F.I.) personally believe the results of Marchetti et al. to be artefactual, caused by receptor overexpression and the deletion construct used in the experiments. My group had already shown in our 2009 Neuron publication (the first one on this topic) that the majority of p75^{NTR} molecules at the cell surface are not disulphide linked after overexpression in cells, thus Marchetti et al. did not really break any new ground in that respect. In contrast, most if not all receptor molecules are disulphide linked in neurons endogenously expressing p75^{NTR}. Moreover, the study was based on a receptor construct bearing a deletion of the juxtamembrane region, which includes a palmitoylation site and binding sites for several downstream molecules, including TRAF6, NRIF, RhoGDI and others. Deletion of this region will most certainly alter the affinity and dynamics of the way the receptor interacts with ligands and downstream pathways, affecting signaling and trafficking in unpredictable ways. In many receptors, that kind of deletion causes constitutive and/or promiscuous activation. I also note that it was shown in various papers using different methods that the complex between the neurotrophin dimer and p75^{NTR} involves a dimer, not a monomer, of receptor subunits (namely a 2:2 complex). NMR structures of the p75^{NTR} TMD are now available from 3 independent research groups, and all show it to form dimers. The p75^{NTR} death domain also forms low-affinity dimers in solution. There is however one paper (and only one) published in *Science* in 2004 by Dr. Garcia and colleagues claiming that the NGF:p75^{NTR} complex involves only one receptor molecule bound to the NGF dimer (i.e. a 2:1 complex). This result was later proven by other groups to be an artefact caused by the artificial de-glycosylation of the receptor molecules used to produce the crystal structure reported by Garcia et al. Indeed, by both x-ray crystallography (Gong et al. Nature 2008) and analytical ultracentrifugation (Aurikko et al. JBC, 2005), the stoichiometry of the NGF:p75^{NTR} complex has been proven to be 2:2, namely a receptor dimer bound to a neurotrophin dimer. In

conclusion, I do not have any confidence in the results presented by Marchetti et al. A proper discussion of the issues raised by that paper is beyond the scope of the present manuscript and I therefore respectfully decline to refer to it here.

b. Also on page 4, the authors imply that the Cys-mutant in the TM of p75 impairs p75 action in an AD mouse model- but in fact, this work suggests that what is affected is trafficking of Abeta, not p75 signaling.

We respectfully disagree with this statement. The reviewer is here alluding to our 2021 EMBO Journal paper by Yi et al. that used the 5xFAD mouse model. The sentence in page 4 where this reference appears reads as follows: “*Knock-in mice carrying this replacement show significant protection from epileptic-induced neuronal damage (Tanaka et al, 2016) as well as neuropathology and memory impairment caused by over-expression of the 5xFAD Alzheimer’s disease transgene (Yi et al, 2021).*” We fail to see how this sentence conveys the notion stated by the reviewer.

We take the opportunity to clarify that the inability of the TMD Cys mutant of p75^{NTR} to signal in response to neurotrophins has been proven numerous times, including in our first 2009 *Neuron* paper on this topic. In that work, we showed that NGF fails to activate a number of effectors and pathways, including NF- κ B, JNK, TRAF6, NRIF and caspase-3 through the Cys mutant receptor. In a subsequent publication (Tanaka et al. 2016) we showed that knock-in neurons endogenously expressing this mutant failed to respond to proneurotrophins. Moreover, our paper by Yi et al. shows that the C259A p75^{NTR} mutant (endogenously expressed in neurons from knock-in mice) has signaling impairments in response to Abeta as well as trafficking deficits (i.e. enhanced recycling in Rab11 endosomes to the plasma membrane) which arise from its inability to signal, as it is well known that most receptors escape the recycling route and internalize to Rab5/Rab7 endosomes when activated. Due to the ability of p75^{NTR} to directly interact with APP, this in turn affects the internalization and trafficking of APP (not “Abeta”, as stated by the reviewer), and as a consequence A β production. From all this we conclude that the Cys mutation did indeed affect p75^{NTR} action in the 5xFAD model.

c. On page 9, the authors refer to a melanoma invasion model, but this is simply an in vitro migration model- this should be corrected in Fig 3 and in Fig 6 and in the text that refers to both figures.

This is a semantic issue. The cells migrate and thus invade the lower compartment of the chamber. Cells can also move and migrate on the surface of the upper compartment but that would not be called invasion. This is accepted wording often used in the literature when referring to the Boyden chamber assay.

d. The authors use A875, A875-WT, A875-NT seemingly interchangeably. Is the A875-NT a real control (ie scrambled shRNA) or is it just wildtype cells.

As described in our previous paper involving these cells (Goh et al. 2018), the PiggyBac non-targeting (NT) vector contains a shRNA insert (sequence; CAACAAGATGAAGAGCACCAA, provided by the manufacturer System Bioscience) that does not target any known genes from any species. We have now clarified this also in this manuscript.

Reviewer #2

1. ***In Figure 3B, wound healing assay showed that A875 shp75 cells is as invasive as A875 NT cells that both wound areas decreased from 1 to ~0.75 in 24 hours. But in Figure 8C, A875 shp75 showed significant less metastasis in the lungs in SCID mice. Do the in vitro assays recapitulate melanoma invasiveness? If not, is it possible the ideal compounds were missed from the screening in melanoma cells (Figure 6)?***

We would like to clarify that the screening of compounds was not done in melanoma cells but with the AraTM assay in bacteria. This assay simply scores the relative orientation/conformation of the interacting receptor TMDs in isolation from the rest of the receptor. The wound healing assay is a rather simplistic assay of cell motility. The in vitro assay that better recapitulates what may happen in vivo is the Boyden chamber migration/invasion assay.

2. ***In Figure 7A, co-IP experiment requires input and IgG control to show positivity and specificity. In Figure 7B, quantification using the ratio of Triton-insoluble fascin to total fascin will be much more solid than only showing the western blot.***

The input control was added in 7A and quantification in 7B (also in 7A).

3. ***Also in Figure 7A, the labelling of NGF +/- does not match what is described in the text "NGF treatment strongly stimulated the association of fascin with p75NTR".***

The labels in the Western blot shown in this panel were incorrect. They have now been amended.

4. ***Validation of results in another human cell line***

During the past year, multiple tests were done by intravenous injection of human 70W cells, which showed moderate p75^{ntr} expression in vitro as assessed by Western blotting, followed by monitoring cells in the lungs in the presence and absence of drug treatment. Unexpectedly, and unlike A875 cells, we found that 70W cells lose p75^{ntr} expression in the in vivo environment. Although they metastasized in large numbers, very few 70W cells expressed little, if any, p75^{ntr} in the lung. The fact that 70W cells could form lung tumors without expressing p75^{ntr} indicates that these cells do not depend on this receptor for their motility and tissue invasion, and probably use alternative pathways. Unsurprisingly, drug treatment had no effect on the number of metastasized cells.

To circumvent this problem and assess the ability of Np75-4A22 to inhibit migration/invasion in melanoma cell lines other than A875, we tested the compound in vitro on the Boyden chamber assay using the WM115 human melanoma cell line, which expresses p75^{NTR} at levels intermediate between A875 and 70W cells. As shown in Figure 6E of our revised manuscript, NGF induced the migration of these cells to the lower side of Boyden chambers and, importantly, 1 μ M Np75-4A22 blocked this activity without affecting the spontaneous migration of the cells, suggesting a broad ability to prevent invasion of human melanomas that express p75^{NTR}.

We also note that, as advised by the editor, we no longer refer to the effects of the compounds as affecting melanoma metastasis, but instead cell migration, motility or invasion as appropriate.

Reviewer #3

- 1. A series of in vitro experiments indicated that the Np75-4A22 did not induce apoptosis of melanoma cells while Div17E5 did, but was more potent at blocking melanoma cell motility. Figure 8 shows an in vivo study demonstrating that Np75-4A22 reduced A875 melanoma cells using a mouse xenograft model. Although the in vitro studies showed solid evidence that Np75-4A22 has more potent to block cell motility compared to Div17E5, it is not clear about its advantage in in vivo study over Div17E5. The data shown in Figure 8 indicated that it reduced the tumor growth of melanoma cells. However, the same result probably could be obtained from the same experiments using pro-apoptotic chemicals. To support the idea that the activity of blocking cell motility of Np75-4A22 would be superior to Div17E5, I believe some additional data are needed. e.g., Showing the same type of in vivo data with Div17E5 and seeing whether Np75-4A22 is more effective. Also, in Figure 8C, the authors compared the number of cancer cells among the conditions, but if they emphasized the anti-migration/metastasis function of Np75-4A22, it would be interesting whether there is a difference in the number of cancer lesions between these conditions. If the number of lesions in the condition with Np75-4A22 is less than that in Div17E5, it would be another evidence to concrete their insistence.**

While both Div17E5 and Np75-4A22 showed effects on cell migration/invasion in vitro, Div17E5, but not Np75-4A22, induced melanoma apoptosis in culture. In this study, our goal was to assess the ability of small molecules targeting the TMD of p75^{NTR} to interfere with melanoma migration, invasion and metastasis. Hence the use of Np75-4A22, and not Div17E5, in the in vivo study. The inability of Np75-4A22 to induce melanoma apoptosis in vitro makes it unlikely that it affected cell migration/invasion in vivo by killing the melanoma cells. Our previous paper (Goh et al. 2018) already showed that a small molecule targeting the TMD of p75^{NTR} can induce melanoma apoptosis and interfere with the growth of melanoma tumors on the flank of mice.

The number of cancer lesions has been added to this figure.

- 2. The authors think that one of the mechanisms that Np75-4A22 impairs NGF-mediated melanoma migration is via reducing fascin/p75NTR interaction. Because fascin is an actin-bundling protein, they should examine if Np75-4A22 treatment cause the structural changes of actin cytoskeleton such as lamellipodia formation or filopodia formation. The labeling in figure 7A seems to be wrong. Please check and correct it.**
The effects of NGF in the presence and absence of Np75-4A22 on the actin cytoskeleton of melanoma cells have been investigated and the results added to the revised manuscript (Figure 7C-E). A significant increase in the number of filopodia per cell was observed upon NGF treatment of A875 NT cells which was not detected in A875 shp75 cells, indicating NGF-induced effects on the actin cytoskeleton that were mediated by p75^{NTR}. Np75-4A22, at either 1 or 5 μ M, had no effects per se in either cell line but completely abolished the effects of NGF on filopodia formation in A875 NT cells. These results indicated that Np75-4A22 may inhibit melanoma cell migration and invasion by interfering with NGF-mediated filopodia formation.
- 3. One weakness of this manuscript is the unclearness of the statistical analyses. The method section mentions that the data were shown as mean{plus minus}SEM, and statistical significance was calculated using the Student t-test, Mann Whitney, or one-way ANOVA. However, in many of the figures with statistical analyses, there is no description of which statistical test was used. The methods should be described in each legend. Also, if they used the Mann-Whitney test, I assume some of the data distribution was non-parametric. If so, they should use other ways to display the data, such as the box-and-whisker plot. Using mean{plus minus}SEM for non-parametric**

data often causes misleading.

Student t-test was only used for the graph presented in Figure 7B. This is indicated in the corresponding figure legend. For all other data one-way ANOVA followed by Tukey's multiple comparisons test was used. This has been stated in the Methods section which now reads as follows "Statistical significance was calculated using one-way ANOVA followed by Tukey's multiple comparisons test for all data except that shown in Figure 7B which used the Student t-test". Mann Whitney was incorrectly mentioned in the Methods section which has been corrected and amended to clarify this.

- 4. Another weakness is the biological specificity of using p75NTR as a drug target. It has been well known that p75NTR is expressed in many normal cell types through various organs. They mention, 'No mortality or adverse clinical signs, including body weight, food consumption or effects on gross histological pathology assessed on formalin fixed paraffin embedded lung, liver, kidney or brain were observed in any of the Np75-4A22 treated mice. However, in the brain, it has also been known that p75NTR has an important role in memory formation, which regular histological pathology cannot assess. Although I don't necessarily think they should add such data, it would be better to discuss it as a potential side effect of the compound.'***

In the Discussion, we have added the following sentence: "We cannot rule out at this point that the treatment with Np75-4A22 affected brain functions which standard histological pathology cannot assess, such as effects on learning or memory."

- 5. The authors used the AraTM transcription factor system in bacteria for chemical compound screening and identified 4 potential compounds (Figure 1A). However, it was unclear how they further narrowed it down to the one (Div17E5). In the text, they mention 'and subjected to secondary and tertiary screens for reduced toxicity and increased potency and specificity.' I could not figure out what assays were used for these additional screenings. During the screening procedure, it is very important to narrow down to the small number from many candidates. It would be useful information for the readers on how they perform the secondary and tertiary screens.***

From Figure 1A it can be seen that compound Div17E5 was the one that elicited the strongest fold change in GFP/OD630 from the 4 potential compounds labeled in red. Reduced toxicity refers to the OD630 reading, increased potency refers to the GFP/OD630 response and specificity refers to absence of (or negligible) effects on unrelated TMDs such as that from $\alpha 2\beta 3$ integrin, as performed in our previous study (Goh et al. 2018). This has now been clarified in the corresponding section.

Minor points:

Figure 1:

Panel names in the main text and the actual panels are not consistent. e.g. p6. 18 and 19, Figure 1D, E should be 1D, 1F, and p6. 19, 1F should be 1E. Figure notes in both 1G and 1H use colored squares, but the figures use a square and a circle. In the text, they say, 'The compound induced oscillations in real-time anisotropy (Figure 1G)'; however, the figure seems that the vehicle and Div17E5 were flipped.

All these errors have been corrected in the revised version of the manuscript.

Figure 2:

The order of figure legends is inconsistent with the text and figures. Please correct them.

The legend to this figure has been corrected in the revised version of the manuscript.

Figure 3:

I believe the lower panel of current B should be C. C should be D, and D should be E. In the figure legend for (A), it is not clear what 'Size bar, Xmm' means. It is probably better to add references for p7. 113 ('As p75NTR is known to regulate motility and migration in melanoma cells').

This has been corrected in the revised version of the manuscript.

Figure 4:

p8. 120, the authors claim, 'However, when tested in the wound healing assay in A875-NT cells, both compounds were less potent than Div17E5.' There is no statistical analysis on this panel (4D), so the overstatement should be avoided. If they want to clearly state 'were less potent,' then the statistical analysis, such as two-way ANOVA, should be added.

This sentence has been changed to “However, when tested in the wound healing assay in A875-NT cells, Div17E5 was still the better of the three (Figure 4D).”

Figure 5:

Panel B has no asterisks for the stats.

The asterisks were added in this panel.

Figure 6:

Figure 6A, as the conditions are defined quantitatively, it would be more informative if they could present quantitative analysis for the WB signals (this can also be possible for Figure 2B). Since it is described that the experiment was repeated three times with comparable results, this is feasible. The note for Figure 6B (bottom) misses symbols for A875 shp75 + drug and A875 shp75 + vehicle.

Quantification of results in both these figures are now shown in the revised version. Note for the symbols in Figure 6B has been amended.

6th Jun 2025

Dear Prof. Ibanez,

Thank you for submitting your revised study.

Referee #3 has reviewed your revised manuscript and your rebuttal to all referees' concerns, and as you will see below, is satisfied with the revisions. I will therefore be able to accept your manuscript once the following editorial issues are addressed:

1/ Manuscript text:

- Please indicate in track changes mode any new modification in the text.
- There is a name discrepancy between the manuscript and the submission system: Vanessa Lopes-Rodrigues vs. Vanessa Rodrigues; please check and correct where needed.
- Authors Xueyan Han, Liane Babes, Angela Z. Chou, Lingyu Du, Siyi Dong, James J. Chou are not entered in the submission system.
- We can accommodate a maximum of 5 keywords, please adjust accordingly.
- Please remove "not shown" (p.8). As per journal policy, all results discussed in the manuscript must be shown in the main or supplementary figures.
- Methods:
 - o Please download and fill our Reagents and Tools Table template (.docx), which you can find in our author guidelines: <https://www.embopress.org/page/journal/14693178/authorguide#structuredmethods>.
 - o Cells: please provide the origin and indicate whether cells were authenticated and tested for mycoplasma contamination.
 - o Statistics: please provide a statement on blinding and randomization, as well as on sample size and inclusion/exclusion criteria.
 - o There is a supplementary file with text, should it be part of the Methods?
- Acknowledgements: the information provided in the manuscript should match the information provided in the submission system. Currently, this information is missing in the submission system.
- Author contributions: CRediT has replaced the traditional author contributions section because it offers a systematic machine readable author contributions format that allows for more effective research assessment. Please remove the Authors Contributions from the manuscript and use the free text boxes beneath each contributing author's name in our system to add specific details on the author's contribution. More information is available in our guide to authors.
- Disclosure statement and competing interests: We updated our journal's competing interests policy in January 2022 and request authors to consider both actual and perceived competing interests. Please review the policy <https://www.embopress.org/competing-interests> and update your competing interests if necessary.

2/ Figures:

- Please provide individual production quality figure files as .eps, .tif, .jpg (one file per figure). For guidance, download the 'Figure Guide PDF' (<https://www.embopress.org/page/journal/17574684/authorguide#figureformat>).
 - Please provide exact p values in the figures or in their legends.
 - Please provide individual data points where possible.
 - Please make sure all figures/figure panels are referenced in the text (currently a callout is missing for Fig. 8F).
 - We replaced Supplementary Information with Expanded View (EV) Figures and Tables that are collapsible/expandable online. EV Figures should be cited as 'Figure EV1, Figure EV2" etc... in the text and their respective legends should be included in the main text after the legends of regular figures.
 - For the figures that you do NOT wish to display as Expanded View figures, they should be bundled together with their legends in a single PDF file called *Appendix*, which should start with a short Table of Content. Appendix figures should be referred to in the main text as: "Appendix Figure S1, Appendix Figure S2" etc.
 - Additional Tables/Datasets should be labeled and referred to as Table EV1, Dataset EV1, etc. Legends have to be provided in a separate tab in case of .xls files. Alternatively, the legend can be supplied as a separate text file (README) and zipped together with the Table/Dataset file.
- See detailed instructions here:

- Please address the queries from our copy editors in the figure legends:

1. Please indicate the statistical test used for data analysis in the legends of figures 1A, D-F; 2D, 3B, D, E; 5A, C; 6B-E; 7A, B, D, E; 8B, C, D, F
2. Please note that information related to n is missing in the legend of figure 1A

3/ Please provide source data for the main figures together with the completed Source Data checklist.

4/ A complete author checklist, which you can download from our author guidelines

(<https://www.embopress.org/page/journal/17574684/authorguide#submissionofrevisions>). Please insert information in the checklist that is also reflected in the manuscript. The completed author checklist will also be part of the RPF.

5/ Please provide 'The paper explained': EMBO Molecular Medicine articles are accompanied by a summary of the articles to emphasize the major findings in the paper and their medical implications for the non-specialist reader. Please provide a draft summary of your article highlighting

6/ Every published paper now includes a 'Synopsis' to further enhance discoverability. Synopses are displayed on the journal webpage and are freely accessible to all readers. They include a short stand first (maximum of 300 characters, including space) as well as 2-5 one-sentences bullet points that summarizes the paper. Please write the bullet points to summarize the key NEW findings. They should be designed to be complementary to the abstract - i.e. not repeat the same text. We encourage inclusion of key acronyms and quantitative information (maximum of 30 words / bullet point). Please use the passive voice. Please attach these in a separate file or send them by email, we will incorporate them accordingly.

Please also suggest a visual abstract to illustrate your article as a PNG file 550 px wide x 300-600 px high. A cropped portion of this image will serve as thumbnail for the table of content on our webpage.

7/ As part of the EMBO Publications transparent editorial process initiative (see our Editorial at <http://embomolmed.embopress.org/content/2/9/329>), EMBO Molecular Medicine will publish online a Review Process File (RPF) to accompany accepted manuscripts.

This file will be published in conjunction with your paper and will include the anonymous referee reports, your point-by-point response and all pertinent correspondence relating to the manuscript. Let us know whether you agree with the publication of the RPF.

I look forward to receiving your revised manuscript.

Yours sincerely,

Lise Roth

***** Reviewer's comments *****

Referee #3 (Remarks for Author):

The manuscript was appropriately revised according to my comments and suggestions. I think it is now acceptable for publication in EMBO molecular medicine.

To comments of Reviewer#1

The aim of this study is the identification of a novel chemical compound interacting with p75NTR and to explore its function in cancer biology. I agree with this reviewer that 'Point 1', regarding the structure-function relationship of Np75-4A22 (and Div175), raises an important and intellectually interesting question. However, as the authors reasonably argue in their rebuttal, this question is a curiosity-driven inquiry and beyond the scope of this paper. While the authors did not provide a direct answer to this question, they added data from 2D NMR to show the interaction of Np75-4A22 and p75NTR TMD along with the identification of residues involved in drug binding, which partially addresses this in a meaningful way.

Regarding 'Point 4', the reviewer suggested using a primary tumor xenograft model to assess tumor growth and distant

metastasis. While this would indeed strengthen the study, it is also recognized that intravenous (i.v.) injection models are widely used and accepted in the field. The authors attempted a flank tumor model but found it unsuitable for further analysis. In light of this and given their revised interpretation, which focuses on cell motility and invasion rather than metastasis, I believe their rationale and the inclusion of additional in vitro data using another melanoma cell line are acceptable.

The remaining issues raised by Reviewer#1 are the discrepancy of figure panels and their legends or descriptions in the main text or terminology issues. Most of these have been corrected, and the authors' explanations are convincing. However, for 'Point 5d', the usage of the term A875-WT' remains in the 'Cell culture and immunoblotting' in the methods. Regarding '5c', I have cross-checked relevant literature and confirm that describing the Boyden chamber assay as assessing 'invasion' is commonly accepted.

To comments of Reviewer #2

The authors clarified the confusion related to 'Point 1' and also corrected errors indicated in 'Points 2 and 3'.

For 'Point 4' requesting the data with another cell line, while they did not provide in vivo data due to the unexpected loss of p75 NTR expression in the 70W human melanoma cell line in vivo, they instead included new in vitro data using this line.

Overall, the authors have corrected noted errors and discrepancies between figures and text. For the points they did not directly address with new experiments, their justifications are reasonable and convincing.

The authors addressed the remaining reviewer's comments.

1st Jul 2025

Dear Prof. Ibanez,

Thank you for submitting your revised study, and please accept my apologies for the delay in getting back to you as I have been out of the office recently and we have experienced a large number of submissions.

I have now gone through your files, and before I can accept your manuscript, the following editorial matters must be addressed:

1/ Manuscript text:

- Please accept previous changes and only keep in track changes mode any new modification in the text.
- Thank you for removing 'not shown', but the data are still mentioned. Either provide the data or remove the text referring to it.
- Methods:
 - o Please provide dilutions/concentrations for all antibodies.
 - o Animals: please provide age, housing and husbandry conditions, and origin of the C57BL/6 mice. Please state details of authority granting ethics approval and provide reference number for approval (including for SCID mice). Include a statement of compliance with ethical regulations.
 - Data Availability: Please note that NMR structures should be deposited in an appropriate repository. Please refer to our guidelines: <https://www.embopress.org/page/journal/17574684/authorguide#datadeposition>

2/ Figures:

- Please provide exact p values in the figures or in their legends.
- Please provide individual data points in your graphs.
- As part of our standard image integrity checks, we identified potential irregularities in your figure set. We would appreciate your assistance in clarifying these issues before we proceed further. Specifically, we ask you to review the following:
 - Figure 2C: The blots appear over-contrasted, with no visible background detail. This issue is also present in the provided source data.
 - Figure 6A: The image appears over-pixelated, and again, no background information is visible. This is also reflected in the source data.
 - Figure 7C: There are obscured black boxes in the figure. Notably, these are not visible when viewing the image without filters.

To address these concerns, we kindly request the following:

*Please provide an unedited version of Figure 7C.

*Please submit updated versions of Figures 2C and 6A, using the original captured 16-bit blot data to ensure full image integrity and background detail.

Please refer to the attached images for reference.

3/ Thank you for providing Source Data. Please note the following:

- Source Data are missing for Figures 3E, 7B (WB), 7C, 7D and 7E.
- The labeling of the tabs for Fig 5 doesn't match the figure.
- In the excel files, please provide individual data points, not means.

4/ Checklist:

- You have filled in the following subsections, please check if they really apply to your study: Primary cultures / Animals observed in or captured from the field
- Please fill in the full section "Experimental study design and statistics".
- Please fill in the Data Availability section/Primary datasets.

6/ I introduced minor edits in your synopsis, please let me know if you agree or amend as you see fit:

"A novel small molecule, Np75-4A22, which targets the transmembrane domain of the p75NTR death receptor, was found to inhibit the migration and lung invasion of human melanoma cells in mice.

- The specific interaction of Np75-4A22 with the p75NTR transmembrane domain was confirmed by 2D NMR.
- Np75-4A22 antagonized NGF-mediated recruitment of the actin-bundling protein fascin to p75NTR, as well as fascin association with the actin cytoskeleton and filopodia formation.
- Np75-4A22 displayed high oral bioavailability, low toxicity, and inhibited melanoma lung invasion in mice."

Thank you for providing a nice visual abstract. Please reduce its size to 550 px wide x 300-600 px high, and make sure that the text remains legible. I have cropped the attached image to serve as thumbnail for the table of content on our webpage. Please let me know if you agree, or provide an alternative image at the right dimensions (115px x 70px).

7/ As part of the EMBO Publications transparent editorial process initiative (see our Editorial at

<http://embomolmed.embopress.org/content/2/9/329>), EMBO Molecular Medicine will publish online a Review Process File (RPF) to accompany accepted manuscripts. We note that you agree with the publication of the RPF.

Looking forward to hearing from you.

With kind regards,

Lise Roth

The authors addressed the editorial issues.

1st Aug 2025

Dear Prof. Ibanez,

Thank you for providing the revised files and for your patience while we went through all of them.

Almost everything is fine now, and I should be able to accept your manuscript once the following remaining editorial issues are addressed:

1. In the methods, please provide the dilutions/concentrations for all antibodies, including immunofluorescence and immunoblots.

2. Fig 4D and 6D contain graphs with n=2; please remove the error bars or justify.

3. In your source data:

a. Files with individual data points are missing for Fig. 3B, C, 6B, C.

b. Raw / unprocessed data is missing for Fig. 6E, 7B (WB), 7C, 7D, 7E.

4. Synopsis:

I introduced minor edits in your synopsis, please let me know if you agree or amend as you see fit:

"A novel small molecule, Np75-4A22, which targets the transmembrane domain of the p75NTR death receptor, was found to inhibit the migration and lung invasion of human melanoma cells in mice.

- The specific interaction of Np75-4A22 with the p75NTR transmembrane domain was confirmed by 2D NMR.

- Np75-4A22 antagonized NGF-mediated recruitment of the actin-bundling protein fascin to p75NTR, as well as fascin association with the actin cytoskeleton and filopodia formation.

- Np75-4A22 displayed high oral bioavailability, low toxicity, and inhibited melanoma lung invasion in mice."

Please reduce the size of the visual abstract to 550 px wide x 300-600 px high, and make sure that the text remains legible. Please also let us know whether you agree with the cropped selection to serve as a thumbnail (attached).

Looking forward to hearing from you.

With kind regards,

Lise Roth

Lise Roth, PhD

Senior Editor

EMBO Molecular Medicine

The authors addressed the remaining editorial issues.

11th Aug 2025

Dear Prof. Ibanez,

Thank you for your patience and for submitting the last revised files. I am pleased to inform you that your manuscript is accepted for publication and is now being sent to our publisher to be included in the next available issue of EMBO Molecular Medicine!

Please check Figure 6 again, as I'm not sure the panels have been updated. If changes need to be made, please let us know immediately.

Yours sincerely,
